# Capturing COVID-19 spread and interplay with multi-hop contact tracing intervention

**Jungyeol Kim** [ORCID]*, **Shirin Saeedi Bidokhti, Saswati Sarkar**

Department of Electrical and Systems Engineering, University of Pennsylvania, Philadelphia, PA, United States of America

* jungyeol@seas.upenn.edu

## Abstract

A preemptive multi-hop contact tracing scheme that tracks not only the direct contacts of those who tested positive for COVID-19, but also secondary or tertiary contacts has been proposed and deployed in practice with some success. We propose a mathematical methodology for evaluating this preemptive contact tracing strategy that combines the contact tracing dynamics and the virus transmission mechanism in a single framework using microscopic Markov Chain approach (MMCA). We perform Monte Carlo (MC) simulations to validate our model and show that the output of our model provides a reasonable match with the result of MC simulations. Utilizing the formulation under a human contact network generated from real-world data, we show that the cost-benefit tradeoff can be significantly enhanced through an implementation of the multi-hop contact tracing as compared to traditional contact tracing. We further shed light on the mechanisms behind the effectiveness of the multi-hop testing strategy using the framework. We show that our mathematical framework allows significantly faster computation of key attributes for multi-hop contact tracing as compared to MC simulations. This in turn enables the investigation of these attributes for large contact networks, and constitutes a significant strength of our approach as the contact networks that arise in practice are typically large.

## 1 Introduction

Traditional contact tracing is known to be one of the effective ways to control epidemics through quickly identifying and isolating individuals who quietly infect others without displaying symptoms [1, 2]. Thus, public health authorities such as the US Center for Disease Control and Prevention (CDC) have recommended to trace direct contacts of those who tested positive. To break the chain of transmission in a more efficient way, a *multi-hop* contact tracing scheme has been proposed and deployed in practice with some success. The multi-hop contact tracing scheme traces not only the direct contacts of those who tested positive, but also secondary contacts (i.e., contacts of contacts) or tertiary contacts (i.e., contacts of secondary contacts) and even higher order contacts. The first work in this field proposed to test secondary contacts and evaluated it using simulations [3]. Recently we have proposed a generalized multi-hop contact testing which tests k-hop contacts and decides the number k considering the benefit (reduction in outbreak size) and cost (the number of individuals isolated) as a function of k

**Data Availability Statement:** All relevant data are within the manuscript and its Supporting information files.

**Funding:** NSF CAREER award 2047482, NSF grant 1910594, and NSF grant 2008284; The funders

had no role in study design, data collection and analysis, decision to publish, or preparation of the manuscript.

**Competing interests:** The authors have declared that no competing interests exist.

[4]. Our extensive simulations on various contact networks reveal that the tradeoff between cost and benefit can be significantly enhanced through the multi-hop tracing scheme as compared to the traditional contact tracing [4]. In practice, Vietnam has implemented the multi-hop contact tracing scheme, and during the first 100 days of the epidemic, many people who were contact-traced and quarantined turned out to be actually the secondary contacts of those who had tested positive [5].

Given that the state of the art research and deployment in practice promises that multi-hop contact tracing can effectively control the spread of infectious diseases, it is imperative to devise computationally efficient strategies for thorough evaluation. So far the evaluations have only been through simulation. The limitation of simulation approach in realistic scenarios is that it is computationally expensive as it requires a large number of iterative runs for accurately estimating statistics based on the results at any finite time, e.g., how does the type of contact tracing strategy affect the probability of identifying a superspreader in close contact with a large number of people? In this paper we propose an analytical methodology for evaluating multi-hop contact tracing strategy. Towards this end we incorporate the dynamics of virus transmission through a compartmental model and the dynamics of multi-hop contact tracing through a microscopic Markov Chain approach (MMCA) [6]. This combination has been successfully deployed to model a traditional (i.e., 1-hop) contact tracing strategy in a recent study [7]. The formulation introduced in this recent study shows that reducing the number of infections through traditional contact tracing comes at the cost of an increase in the number of individuals isolated [7].

The current paper is the first work to provide a mathematical model for investigating multi-hop contact tracing strategy. We combine the multi-hop contact tracing dynamics and the virus transmission mechanism in a single framework using microscopic Markov chain approach (MMCA) [6]. The framework can compute various attributes such as the number of infections in any desired time period, the number of individuals quarantined, the number of individuals tested, etc. For simplicity of exposition, we present the framework for 2-hop contact tracing in the main body of the paper, and generalize to accommodate $k$-hop contact tracing in the S1 Text. We first perform Monte Carlo (MC) simulations to validate our model and show that the output of our model provides a good match with the result of MC simulations. We then utilize our formulation to show the effectiveness of multi-hop contact tracing scheme in controlling an epidemic for a human contact network generated from real-world data. Utilizing our formulation we show that multi-hop contact tracing can significantly reduce the number of infections, as compared to 1-hop contact tracing, and even while doing so, multi-hop contact tracing decreases the number of people quarantined. Thus multi-hop contact tracing incurs benefits (i.e., reduce the number of infections) while reducing cost (the number of people quarantined), which is apparently counter-intuitive but clearly enhances the cost-benefit tradeoff. We analyze the root cause for enhancing this tradeoff by investigating the connectivity pattern of individuals in the contact networks and the probability of identifying a superspreader with a large number of contacts. Our numerical computations reveal that under multi-hop contact tracing infected nodes have almost zero probability of evading identification and consequently the number of infections can be significantly reduced when multi-hop contact tracing is performed. The number of individuals quarantined reduces because the overall number of infections is significantly reduced. We finally compare the computational time for running the MC simulations with that for obtaining the output of the MMCA formulation. The comparison shows that MMCA formulation allows for significantly faster computation than MC simulations and the computation advantage of the former increases with increase in size of the contact networks. This allows MMCA formulation to efficiently scale to contact networks which arise in practice—

those that arise in practice are invariably large and therefore present significant computation challenges for the MC simulation. Throughout the paper we consider the specific context of COVID-19, though our formulation can be adapted to any other contagious disease that spreads through contact by choosing the appropriate compartmental model for the disease.

## 2 Model formulation

To combine the contact tracing dynamics and the virus transmission mechanism in a single framework, we construct an epidemic model with the following 7 compartments: Susceptible ($S$), Latent ($L$), Presymptomatic ($I_p$), Symptomatic ($I_s$), Asymptomatic ($I_a$), Detected ($D$), and Removed ($R$). Fig 1 illustrates the different stages of the disease and notations representing probabilities of transition between them at any given time.

When a susceptible ($S$) individual comes into contact with a symptomatic ($I_s$) individual, he is infected with the probability of infection $\beta_{I_s}$. Similarly, a presymptomatic ($I_p$) and an asymptomatic ($I_a$) individual infects a susceptible upon contact with transmission probabilities, $\beta_{I_p}$ and $\beta_{I_a}$, respectively. Once an individual is infected he becomes Latent ($L$) in which the individuals never develop symptoms, do not infect others for a mean latency duration of $1/\lambda$, and subsequently become either asymptomatic ($I_a$) with a probability $p_a$ or presymptomatic ($I_p$) with a probability $(1 - p_a)$. An asymptomatic individual remains contagious without displaying symptoms for a mean duration of $1/\mu$, if he isn't diagnosed and removed earlier; after this duration he is no longer contagious, we consider that he is removed ($R$). Likewise, without being detected by a health authority, a presymptomatic individual remains contagious without displaying symptoms yet for a mean duration of $1/\alpha$; after this duration he develops symptoms and is called symptomatic. A symptomatic individual continues to infect his contacts for a mean duration of $1/\mu$ before he moves to removed ($R$) state if he is not detected before.

When an individual tests positive, he is said to be detected and enters the detected (D) state and remains there subsequently. A detected individual is quarantined until he is no longer contagious, i.e., until he either recovers from the disease or is dead. Thus detected individuals can not infect others and are effectively removed from the system. Note that we do not consider re-infection in this paper. We consider that every test is accurate, i.e., there are no false positives and no false negatives. Thus, whenever tested 1) presymptomatic, asymptomatic and symptomatic individuals test positive and are detected 2)

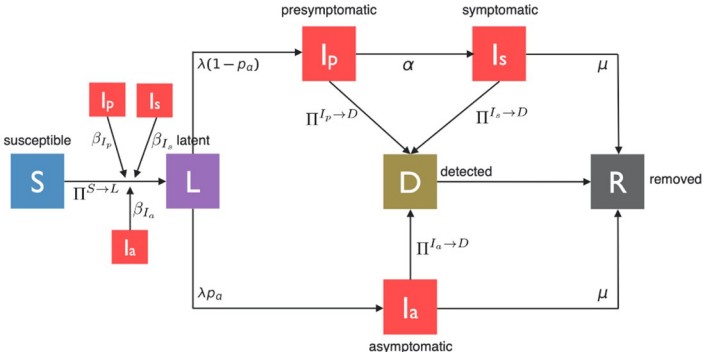

**Fig 1. A pictorial illustration of state transitions involved in the spread and evolution of COVID-19.**

susceptible and latent individuals test negative and are not detected (susceptible individuals test negative because they are not infected and since there is no false negative, latent individuals are already infected but the virus has not grown in them to the point that it can be detected through tests). Presymptomatic and asymptomatic individuals can be tested only as a result of contact tracing as they do not display symptoms. Symptomatic individuals can be tested either 1) after being traced from a contact or 2) because they develop symptoms. We consider that a symptomatic individual is tested because he develops symptoms with probability $\omega$, $\omega$ can be less than 1 as not everyone reports his symptoms.

Contact tracing will be done from any individual detected by a health authority. However, it has been documented that only a fraction of contacts of patients who test positive can be tested, as many contacts either do not consent or can not be obtained or traced by public health authorities [8]. We consider cases in which only $f$ proportion of interactions per day can be identified by a health authority, i.e., each contact (i.e., each edge) information is acquired for contact tracing with probability $f$ each day by a health authority. We refer to $f$ as activation probability. Contact-tracing apps installed in wearable or hand-held devices can increase the amount of information on interactions among people, but still do not guarantee full acquisition of contact information as individuals may not install the apps and may not carry their personal devices all the time. Hence, activation probability $f$ may be low in real life. Finally, a detected ($D$) individual transits to Removed ($R$) state with probability 1, meaning that the detected individual transitions to Removed state the next day. Refer to Table 1 for the disease parameters.

## 2.1 Dynamics of epidemic spreading and contact tracing

The probabilities that a node $i$ is in each compartment at time $t$ are denoted as $\rho_i^S(t)$, $\rho_i^L(t)$, $\rho_i^{I_p}(t)$, $\rho_i^{I_s}(t)$, $\rho_i^{I_a}(t)$, $\rho_i^D(t)$, and $\rho_i^R(t)$ (Susceptible, Latent, Presymptomatic, Symptomatic, Asymptomatic, Detected, and Removed, respectively). The dynamics of spreading of epidemics and different types of contact tracing intervention can be captured by the discrete-time evolution of theses probabilities in the form of microscopic Markov chain approach (MMCA) [6]. The dynamics depend on both time and space, we capture this dependence through explicitly expressing these probabilities as a function of time $t$ and including node identity $i$ as a parameter. We devise a framework to compute these probabilities. The evolution of these probabilities

**Table 1. Disease parameters.**

| $\lambda^{-1}$ | mean latency period |
|---|---|
| $\mu^{-1}$ | mean duration in asymptomatic/symptomatic stage |
| $\alpha^{-1}$ | mean duration in presymptomatic stage |
| $p_a$ | proportion of infections that are asymptomatic |
| $\omega$ | probability that a symptomatic is tested because he shows symptoms |
| $f$ | activation probability; |
| | probability that each contact information (i.e., edge) is acquired for contact tracing by a health authority |
| $\Pi_i^{S \to L}(t)$ | probability that a susceptible node $i$ is infected at time $t$ |
| $\Pi_i^{X \to T}(t)$ | probability that a node $i$ in state $X$ is tested at time $t$ |
| $\Pi_i^{X \to D}(t)$ | probability that a node $i$ in state $X$ is detected via any route at time $t$ |

is given by

$$\rho_i^S(t+1) = [1 - \Pi_i^{S\to L}(t)]\rho_i^S(t)$$

$$\rho_i^L(t+1) = (1-\lambda)\rho_i^L(t) + \Pi_i^{S\to L}(t)\rho_i^S(t)$$

$$\rho_i^{I_p}(t+1) = [1 - \Pi_i^{I_p\to D}(t)](1-\alpha)\rho_i^{I_p}(t) + \lambda(1-p_a)\rho_i^L(t)$$

$$\rho_i^{I_a}(t+1) = [1 - \Pi_i^{I_a\to D}(t)](1-\mu)\rho_i^{I_a}(t) + \lambda p_a\rho_i^L(t)$$

$$\rho_i^{I_s}(t+1) = [1 - \Pi_i^{I_s\to D}(t)](1-\mu)\rho_i^{I_s}(t) + [1 - \Pi_i^{I_p\to D}(t)]\alpha\rho_i^{I_p}(t)$$

$$\rho_i^D(t+1) = \Pi_i^{I_s\to D}(t)\rho_i^{I_s}(t) + \Pi_i^{I_a\to D}(t)\rho_i^{I_a}(t) + \Pi_i^{I_p\to D}(t)\rho_i^{I_p}(t)$$

$$\rho_i^R(t+1) = \rho_i^R(t) + \rho_i^D(t) + \mu[1 - \Pi_i^{I_s\to D}(t)]\rho_i^{I_s}(t) + \mu[1 - \Pi_i^{I_a\to D}(t)]\rho_i^{I_a}(t).$$

Note that $\rho_i^S(t) + \rho_i^L(t) + \rho_i^{I_p}(t) + \rho_i^{I_s}(t) + \rho_i^{I_a}(t) + \rho_i^D(t) + \rho_i^R(t) = 1$ at each time $t$. In the equations above, $\Pi_i^{S\to L}(t)$ is the probability that a susceptible node $i$ is infected (and then transits to Latent state) at time $t$, which can be described as

$$\Pi_i^{S\to L}(t) = 1 - \prod_{j=1}^{N}[1 - A_{ij}\{\beta_{I_p}\rho_j^{I_p}(t) + \beta_{I_a}\rho_j^{I_a}(t) + \beta_{I_s}\rho_j^{I_s}(t)\}], \tag{1}$$

where $A = (A_{jk})$ is the adjacency matrix of the original graphs $G(V, E)$, in which $|V| = N$, and $A_{ij} = 1$ if there exists an edge between node $i$ and node $j$, and $A_{ij} = 0$ otherwise. Nodes $i$ and $j$ are said to be direct neighbors of each other if $A_{ij} = 1$. Note that a node can be infected only through contact with his direct neighbors. The terms in the bracket [. . .] represent the probability that the node $i$ is not infected via contact with $j$. Thus, $\Pi_i^{S\to L}(t)$ represents the probability that the node $i$ is infected by at least one node.

Furthermore, $\Pi_i^{X\to D}(t)$ is the probability that a node $i$ in the state $X$ is detected with testing at time $t$. We start with a sanity check. The above probability when there is no contact tracing and 1-hop contact tracing have been calculated in [7]; our equations in the absence of any contact tracing provide below the same expression as has been obtained by [7] the probability:

$$\Pi_i^{X\to D}(t) = \begin{cases} 0, & X \in \{I_p, I_a\} \\ \omega, & X = I_s, \end{cases} \tag{2}$$

In the absence of contact tracing, only symptomatic individuals can be detected, and that too with probability $\omega$. Thus, $\Pi_i^{X\to D}(t)$ is $\omega$ for symptomatic individuals while it is 0 for individuals who do not display symptoms.

We now proceed to utilize our framework to obtain the above probability when there is traditional 1-hop contact tracing. It is:

$$\Pi_i^{X\to D}(t) = \begin{cases} 1 - \prod_{j=1}^{N}[1 - A_{ij}f\rho_j^D(t)], & X \in \{I_p, I_a\} \\ 1 - (1-\omega)\prod_{j=1}^{N}[1 - A_{ij}f\rho_j^D(t)], & X = I_s. \end{cases} \tag{3}$$

The terms in the bracket [. . .] represent the probability that the node $i$ is not detected due to tracing from his direct neighbor $j$. Thus, $1 - \prod_{j=1}^{N}[. . .]$ represent the probability that the presymptomatic or asymptomatic node $i$ is traced through 1-hop contact tracing. Similarly, $1 - (1-\omega)\prod_{j=1}^{N}[. . .]$ represent the probability that the symptomatic node $i$ is detected either through 1-hop contact tracing or because it shows symptoms.

We next compute the probabilities that a node $i$ is detected under 2-hop contact tracing. For simplicity, we consider only 2-hop in the main body of this paper, but we provide the generalization to arbitrary $k$ hops in S1 Text. The 1-hop neighborhood of node $i$ is defined as the set of direct neighbors of a source node $i$. Denote $G_i^{(1)}(V_i^{(1)}, E_i^{(1)})$ as the sub-graph consisting of a set $V_i^{(1)}$ of nodes (the 1-hop neighborhood of node $i$ as well as node $i$) and a set $E_i^{(1)}$ of edges (all the edges between them). The 2-hop neighborhood of node $i$ is defined as the set of nodes that are reachable from the source node $i$ in 2 hops or fewer. Denote $G_i^{(2)}(V_i^{(2)}, E_i^{(2)})$ as the sub-graph consisting of a set $V_i^{(2)}$ of nodes (the 2-hop neighborhood of node $i$ as well as node $i$) and a set $E_i^{(2)}$ of edges (all the edges between them); see an example of $G_1^{(2)}(V_1^{(2)}, E_1^{(2)})$ in Fig 2a, a sub-graph consisting of 2-hop neighborhood from a node $i = 1$ and edges between them. The challenge is, if $G_i^{(2)}$ is not a tree-like graph, it is very hard to compute the true probability that a node $i$ is detected under 2-hop contact tracing due to the complex correlations among nodes. Thus, in order to compute the approximate probability, we convert the undirected cyclic graph, $G_i^{(2)}(V_i^{(2)}, E_i^{(2)})$, into an undirected acyclic graph, $\overline{G}_i^{(2)}(\overline{V}_i^{(2)}, \overline{E}_i^{(2)})$, through the following approach.

Let $\Delta_i^{(1)}$ denote the set of direct neighbors of the source node $i$ (i.e., $\Delta_i^{(1)} := V_i^{(1)} \backslash V_i^{(0)}$ where $V_i^{(0)} = i$). Let $\Delta_i^{(2)}$ denote the set of nodes at exactly 2-hop from a source node $i$ (i.e., $\Delta_i^{(2)} := V_i^{(2)} \backslash V_i^{(1)}$). We first traverse from the source node $i$ to its direct neighbors, $\Delta_i^{(1)}$. For 2-hop contact tracing, we then traverse from each node of the set $\Delta_i^{(1)}$ to reachable nodes in the set of $\Delta_i^{(1)} \cup \Delta_i^{(2)}$. During this process, we can visit the previously visited nodes again. In the example of Fig 2a, we first traverse from the source node 1 to a set of direct neighbors $\Delta_1^{(1)} = \{2, 3, 4\}$. We then traverse from node $2 \in \Delta_1^{(1)}$ to a set of reachable nodes $\{4, 5, 6, 7\}$ $\in (\Delta^{(1)} \cup \Delta^{(2)})$. Since node 4 is a node that was already visited, we make a copy of node 4 (i.e., shadow node, colored in blue) and append it as a child of 2. We traverse from nodes 2 and 3 through the same process. The Fig 2b shows the acyclic graph $\overline{G}_i^{(2)}(\overline{V}_i^{(2)}, \overline{E}_i^{(2)})$ converted from the original sub-graph $G_i^{(2)}(V_i^{(2)}, E_i^{(2)})$. Note that $\overline{V}_i^{(0)} = V_i^{(0)}$ and $\overline{V}_i^{(1)} = V_i^{(1)}$. Let $\overline{A}_i^{(2)} = (\overline{A}_{i;jk}^{(2)})$ denote the adjacency matrix of $\overline{G}_i^{(2)}$.

Recall that $\Pi_i^{X \to D}(t)$ is the probability that a node $i$ in the state $X$ is detected at time $t$. The theoretical probability, $\Pi_i^{X \to D}(t)$, for 2-hop contact tracing can be calculated using the acyclic graph assumption:

$$\Pi_i^{X \to D}(t) \approx \begin{cases} 1 - \prod_{j \in \overline{V}_i^{(1)} \backslash \overline{V}_i^{(0)}} [1 - \overline{A}_{i;ij}^{(1)} f \rho_j^D(t) - \overline{A}_{i;ij}^{(1)} f \{1 - \rho_j^D(t)\} \{1 - \prod_{k \in \overline{V}_i^{(2)} \backslash \overline{V}_i^{(1)}} (1 - \overline{A}_{i;jk}^{(2)} f \rho_k^D(t))\}], & X \in \{I_p, I_a\} \\ 1 - (1-\omega) \prod_{j \in \overline{V}_i^{(1)} \backslash \overline{V}_i^{(0)}} [1 - \overline{A}_{i;ij}^{(1)} f \rho_j^D(t) - \overline{A}_{i;ij}^{(1)} f \{1 - \rho_j^D(t)\} \{1 - \prod_{k \in \overline{V}_i^{(2)} \backslash \overline{V}_i^{(1)}} (1 - \overline{A}_{i;jk}^{(2)} f \rho_k^D(t))\}], & X = I_s. \end{cases} \quad (4)$$

The terms in the bracket $[\ldots]$ represent the probability that the node $i$ is not detected by any node in the particular branch including node $j$ (e.g., in the graph $\overline{G}_1^{(2)}$ shown in Fig 2b, there are three branches in the network from the node $i = 1$, each containing nodes 2, 3 and 4, respectively). Thus $1 - \prod_{j \in \overline{V}_i^{(2)}} [\ldots]$ represent the probability that the node $i$ is detected by any of the nodes within radius 2 through 2-hop contact tracing. The second term in the bracket represents the probability that node $i$ is detected by node $j$ (1-hop neighbor). The third term represents the probability that node $i$ is not detected by node $j$ (1-hop neighbor) but is detected by any of the node's 2-hop neighbors in the particular branch including node $j$ (e.g., if $i = 1$ and $j = 2$, the third term represents the probability that node $i = 1$ is not

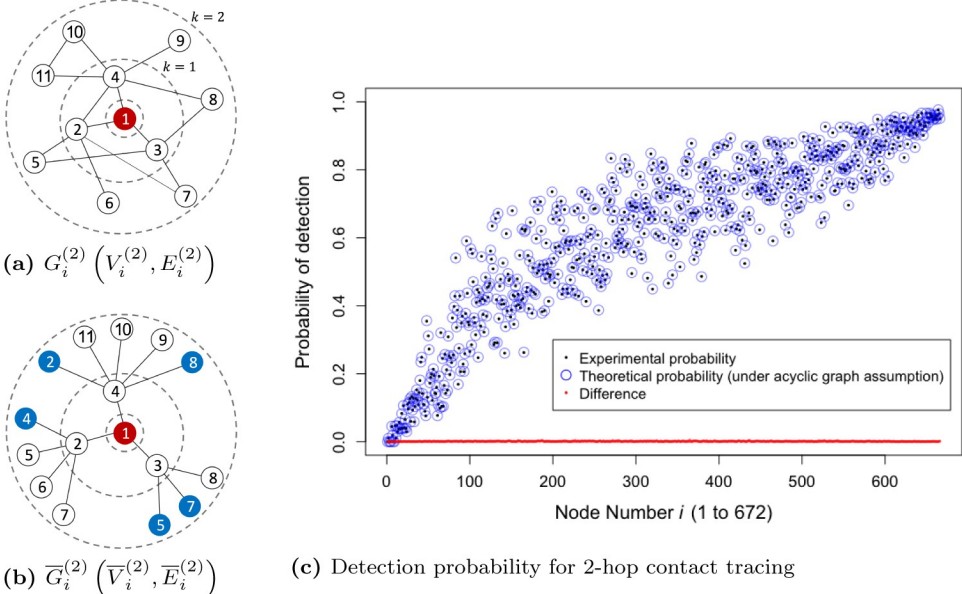

**(a)** $G_i^{(2)}\left(V_i^{(2)}, E_i^{(2)}\right)$

**(b)** $\overline{G}_i^{(2)}\left(\overline{V}_i^{(2)}, \overline{E}_i^{(2)}\right)$

**(c)** Detection probability for 2-hop contact tracing

**Fig 2.** (a) An example sub-graph $G_i^{(2)}$ consisting of 2-hop neighborhood of the source node $i = 1$ and edges between them. (b) A new acyclic graph $\overline{G}_i^{(2)}$ converted from the example sub-graph $G_i^{(2)}$ shown in (a). Shadow nodes are colored in blue. (c) Probability of detection for each node $i$ in University Student Network under 2-hop contact tracing: experimental probability (black dots), theoretical probability under the acyclic graph assumption (blue circles), and the difference between the probabilities. Nodes are numbered in ascending order according to their degree. The theoretical probability, $\Pi_i^{X \to D}(t)$, given in Eq (4) closely approximates the experimental probability; the average of the differences is 0.00098.

detected by node $j = 2$ but is detected by any of the nodes 4, 5, 6, and 7 in the graph $\overline{G}_1^{(2)}$ shown in Fig 2b).

To verify the accuracy of the approximate formulation, for each node $i$, we compare the output of the Eq (4) (under the assumption of acyclic graph, $\overline{G}_i^{(2)}$) to the empirical detection probability obtained from simulation (under the original graph, $G_i^{(2)}$, without the assumption of acyclic graph). We compute the probability of detection for each node $i$ in a data-driven network (a total of 672 nodes and a set of 32, 689 edges) under 2-hop contact tracing scheme with $f = 0.1$ (refer to Section 3 for details on the data-driven network). We assume that a random 1% of the population is in Detected state. The experimental probability without the acyclic assumption is calculated by dividing the number of outcomes with detection by the total number of 100, 000 trials. Fig 2c shows that the output of Eq (4) closely approximates the experimental probability despite the assumption.

Next, we compute the probability that a node $i$ is tested. The probability can be derived from an adaptation of the probability that a node $i$ is detected. Individuals in symptomatic ($I_s$) state can be tested and detected either after showing symptoms or via contact tracing; thus, the probability that an individual in symptomatic ($I_s$) state is tested at time $t$ is equivalent to the probability that the person is detected at time $t$. Moreover, individuals in asymptomatic ($I_a$) and presymptomatic ($I_p$) states can be tested and detected only via contact tracing, and individuals in Susceptible ($S$) and Latent ($L$) state are tested only if they are contact traced by their neighbors, but they do not transit to the detected state; thus, the probability that an individual in those states is tested at time $t$ is equivalent to the probability that the person is detected via

contact tracing at time $t$. Hence, the probability that a node $i$ in state $X$ is tested at time $t$ is

$$\Pi_i^{X \to T}(t) = \begin{cases} \Pi_i^{I_s \to D}(t), & X \in \{I_s\} \\ \Pi_i^{I_a \to D}(t) = \Pi_i^{I_p \to D}(t), & X \in \{I_a, I_p, S, L\}. \end{cases} \tag{5}$$

## 2.2 Evaluation metrics

Using the expressions for the probabilities of states and state transitions obtained in Section 2.1 as a function of time and neighborhood for each node, we define metrics that will be used to quantify the cost-benefit tradeoff of different contact tracing strategies. We first present expressions for the 1) expected number of nodes detected, 2) expected number of nodes detected as a result of contact tracing and expected number of nodes detected as a result of tests initiated because they show symptoms 3) expected number of tests, for the contact tracing strategies we consider. We subsequently obtain expressions for these attributes over time in the time horizon $[1, \ldots, \tau]$ that we consider (Section 2.2.1). We use the above expressions to compare the outbreak sizes, cumulative number of nodes quarantined, tested and detected under the differing contact tracing strategies we consider. When we compare these attributes across contact tracing strategies we distinguish the attributes associated with different contact tracing strategies using the number of hops $k$ of the contact tracing strategies in subscript (Section 2.2.2). Refer to Table 2 for the evaluation metrics.

**2.2.1 Performance metrics for contact tracing strategies.** The expected number of individuals infected during the course of an epidemic by time $\tau$ is

$$I = \sum_{i=1}^{N}[1 - \rho_i^S(\tau)]. \tag{6}$$

The expected number of individuals detected at time $t$ after testing as a result of 1) showing symptoms and 2) contact tracing respectively are:

$$D^S(t) \quad = \omega \sum_{i=1}^{N} \rho_i^{I_s}(t), \tag{7}$$

**Table 2. Evaluation metrics.**

| $\tau$ | Time horizon of evaluation is day 1 to day $\tau$ |
|---|---|
| $I_{k\text{-}hop}$ | total number of individuals infected under $k$-hop contact tracing by time $\tau$ |
| $RI_{k\text{-}hop}$ | $I_{k\text{-}hop}/I_{0\text{-}hop}$; ratio of number of individuals infected under $k$-hop by time $\tau$ to number of individuals infected under 0-hop by time $\tau$ |
| $D_{k\text{-}hop}^S$ | number of individuals detected by time $\tau$ because they tested as a result of experiencing symptoms |
| $D_{k\text{-}hop}^{CT}$ | number of individuals detected via $k$-hop contact tracing by time $\tau$ |
| $D_{k\text{-}hop}$ | $D_{k\text{-}hop}^S + D_{k\text{-}hop}^{CT}$; total number of individuals detected under $k$-hop contact tracing by time $\tau$ (either because they tested 1) as a result of showing symptoms or 2) through $k$-hop contact tracing) |
| $RD_{k\text{-}hop}$ | $D_{k\text{-}hop}/D_{0\text{-}hop}$; ratio of number of individuals detected under $k$-hop by time $\tau$ to total cases detected under 0-hop by time $\tau$ |
| $T_{k\text{-}hop}$ | number of tests under $k$-hop contact tracing by time $\tau$ |
| $i(k)$ | probability that a node of degree $k$ has been infected by time $\tau$ |
| $d(k)$ | probability that a node of degree $k$ has been detected by time $\tau$ |

$$D^{\text{CT}}(t) \quad = \sum_{i=1}^{N} \left[ \Pi_i^{I_a \to D}(t) \rho_i^{I_a}(t) + \Pi_i^{I_p \to D}(t) \rho_i^{I_p}(t) + (\Pi_i^{I_s \to D}(t) - \omega) \rho_i^{I_s}(t) \right]. \tag{8}$$

Thus, by time $\tau$, the expected number of individuals detected is

$$D = D^S + D^{CT} = \sum_{t=1}^{\tau} [D^S(t) + D^{CT}(t)]. \tag{9}$$

The expected number of individuals tested at time $t$ is

$$T(t) = \sum_{i=1}^{N} \left[ \Pi_i^{S \to T}(t) \rho_i^S(t) + \Pi_i^{L \to T}(t) \rho_i^L(t) + \Pi_i^{I_a \to T}(t) \rho_i^{I_a}(t) + \Pi_i^{I_p \to T}(t) \rho_i^{I_p}(t) + \Pi_i^{I_s \to T}(t) \rho_i^{I_s}(t) \right]. \tag{10}$$

By time $\tau$, the expected number of total tests is

$$T = \sum_{t=1}^{\tau} T(t). \tag{11}$$

Thus, the expected number of undetected infections by time $\tau$, is the difference

$$U := I - D.$$

**2.2.2 Notations used to compare performances of *k*-hop contact tracing strategies across *k*.** We now describe the notations used to compare the attributes in Section 2.2.1 across different values of $k$ for $k$-hop contact tracing strategies.

We will compute the ratio of cases infected via $k$-hop contact tracing by time $\tau$ to the cases infected when no contact tracing is performed (i.e., 0-hop contact tracing). This ratio is denoted as $RI_{k\text{-}hop}$ and obtained from (6) as:

$$RI_{k-hop} = \frac{I_{k-hop}}{I_{0-hop}}, \quad k = 1, 2. \tag{12}$$

where the subscript $k$ is used to distinguish the count in (6) for different values of $k$.

We will compute the ratio of the number of individuals quarantined under $k$-hop contact tracing by time $\tau$ to the corresponding number when no contact tracing is performed. Every detected individual is quarantined and vice-versa. This ratio is therefore denoted as $RD_{k\text{-}hop}$ and computed as the number of individuals detected by time $\tau$ under $k$-hop contact tracing to that under 0-hop (i.e., no contact tracing):

$$RD_{k-hop} = \frac{D_{k-hop}}{D_{0-hop}} = \underbrace{\left[ \frac{D_{k-hop}^S}{D_{0-hop}} \right]}_{RD_{k-hop}^S} + \underbrace{\left[ \frac{D_{k-hop}^{CT}}{D_{0-hop}} \right]}_{RD_{k-hop}^{CT}}, \quad k = 1, 2. \tag{13}$$

Again, here the subscript $k$ is used to distinguish the count in (9) for different values of $k$. We divide $RD_{k\text{-}hop}$ into two parts as well, $RD_{k-hop}^S$ and $RD_{k-hop}^{CT}$ to elucidate how symptom-based detection and contact-tracing based detection respectively contribute to $RD_{k\text{-}hop}$. These parts can be computed using the expressions in (7) and (8).

Furthermore, to shed light on the mechanisms behind the effectiveness of 2-hop contact tracing, we analyze the degrees of the nodes that are 1) infected and 2) detected for different

contact tracing strategies. The degrees of the nodes represent the number of contacts of the corresponding individuals. With some overloading of notation we use $k$ to represent different values of degrees of nodes in addition to different values of the number of hops in contact tracing, because it is customary to use $k$ for both degrees and hops in network science. The probability that a node of degree $k$ has been infected by time $\tau$ is

$$i(k) = \frac{1}{N_k} \sum_{i:|k_i=k} \left[ 1 - \rho_i^S(\tau) \right], \tag{14}$$

where $N_k$ is the total number of nodes with degree $k$. The probability that a node of degree $k$ has been detected by time $\tau$ is

$$d(k) = \frac{1}{N_k} \sum_{i:|k_i=k} \sum_{t=1}^{\tau} \left[ \Pi_i^{I_a \to D}(t) \rho_i^{I_a}(t) + \Pi_i^{I_p \to D}(t) \rho_i^{I_p}(t) + \Pi_i^{I_s \to D}(t) \rho_i^{I_s}(t) \right]. \tag{15}$$

In addition, the difference between the two probabilities, $u(k) := i(k) - d(k)$, is used to represent the probability that a node of degree $k$ has been infected but undetected.

We will compare the number of individuals tested by time $\tau$ under $k$-hop contact tracing for $k = 1, 2$, by studying $T_{2\text{-}hop}$, $T_{1\text{-}hop}$ and the ratio $T_{2\text{-}hop}/T_{1\text{-}hop}$, where the subscripts are used to distinguish $T$ obtained from (11) for different $k$-hop contact tracing strategies, $k = 2, 1$.

## 3 Results

Throughout this section, we assume $\beta_{I_s} = \beta_{I_a} = \beta_{I_p}$ and denote all of these as $\beta$.

### 3.1 Empirical validation

In this section, we validate our model through a comparison between results of Monte Carlo (MC) simulations and MMCA formulation. Considering both the data-driven network we used in Section 2, and synthetic networks (specifically Watts-Strogatz network [9] and scale-free network [10]), we show that the computations from our model equations relatively closely match the results of MC simulations. The data-driven network we consider is generated from data collected by smartphones of University students, as part of the Copenhagen Networks Study [11]. The smartphones were equipped with Bluetooth cards which recorded proximity between participating students at 5-minute resolution. According to the definition of *close contact* by CDC [12], we only used proximity events between individuals that lasted more than 15 minutes in a row in the same day to construct daily contact networks. We postulated that two individuals had a contact if there are at least three consecutive proximity events with a 5-minute resolution between them. The constructed contact network has 32, 689 contacts among 672 individuals. The advantage of this data set is that it provides actual proximity events of a moderate number of individuals. The synthetic networks we consider have $N = 1, 000$ nodes and average degrees of $\langle k \rangle = 8$.

We now describe the synthetic networks we consider in this section. Watts-Strogatz network is a synthetic network generated as follows. We start with a regular ring lattice and each endpoint of each edge is rewired with a certain probability to another node chosen uniformly randomly over the rest of the ring. We choose the Watts-Strogatz as representative of synthetic networks because two important attributes that determine the rate of spread of infectious diseases in contact networks, namely average path lengths, and clustering coefficients, can be controlled in this class through the choice of a parameter, namely the rewiring probability. Accordingly, this class of networks has been utilized extensively for simulation studies in spread and control of infectious diseases over contact networks [13]. Note that the average

path length is defined as follows for a given network. The distance between a pair of nodes is the length of the shortest path between them, i.e., the number of links in the path between them that has the minimum number of links among all paths between them. The average path length for a given network is the average of this distance over all pairs of nodes in it. Clustering coefficient is the average of $C_i$ over all nodes $i$ where $C_i$ is defined as the ratio between the actual number of links between the neighbors of $i$ and the maximum possible number of links between the neighbors of $i$ [9]. As the rewiring probability increases from 0 to 1, 1) average path lengths range from linear to logarithmic functions of the number of nodes 2) clustering coefficients range from high to vanishingly small [9, 14]. The special case of the Watts-Strogatz model in which the rewiring probability is 1 corresponds to a variant of the Erdős Rényi random networks which has also been extensively utilized in the study of spread and control of infectious diseases [15, 16]; we consider this variant as well. Furthermore, we consider scale-free network (generated by Barabási-Albert model) to capture the existence of hubs with an excessive number of connections. Unlike scale-free network, the probability of the existence of the hubs in Watts–Strogatz network is low.

We first compare the total infection counts over the course of 1 year (i.e., $\tau = 365$) obtained from MMCA formulation with an average of 200 runs of the MC simulations under the data-driven network and the Erdős Rényi random networks. We consider different values of the activation probability $f$ under 1-hop and 2-hop contact tracing and focus on average and maximum percentage discrepancies or errors. We assume that a random 1% of the population is initially infected and is in a Latent state. Fig 3 represents the total infection count as a function of transmission probability $\beta$ under 1-hop contact tracing for various values of $f$. This shows that MMCA formulation closely approximate the MC simulation results under 1-hop contact tracing. The trend of the variation of the total infection count as a function of $\beta$ is the same for both MMCA and MC. We compare the results of MMCA and MC for a total of 201 values of $\beta$, varying from 0 to 1 with uniform interval 0.005 (with the exception of replacing 0 with 0.001). We compute the discrepancy, i.e., $\frac{|\text{MMCA}-\text{MC}|}{\text{MMCA}}$, for each value of $\beta$, and average the discrepancies over all the values of $\beta$. For data-driven network, the average discrepancy between the two is (a) 0.1% for $f = 0.3$, (b) 0.2% for $f = 0.5$, and (c) 0.4% for $f = 0.9$; for random network, the average discrepancy is (d) 1.5% for $f = 0.3$, (e) 1.5% for $f = 0.5$, and (f) 2.9% for $f = 0.9$; for scale-free network, the average discrepancy between the two is (g) 1.8% for $f = 0.3$, (h) 2.2% for $f = 0.5$, and (i) 3.4% for $f = 0.9$. The average discrepancy is least for data-driven network, next lowest for random network and highest for scale-free network. All the average discrepancies are nonetheless low, below 3.5%.

Fig 4 shows a comparison between MMCA formulation and MC simulation under 2-hop contact tracing. MMCA still approximates the MC simulation reasonably well. Again the trend of the variation of the total infection count as a function of $\beta$ is the same for both MMCA and MC. For data-driven network, the average discrepancy between the two is (a) 0.2% for $f = 0.3$, (b) 0.8% for $f = 0.5$, and (c) 1.5% for $f = 0.9$; for random network, the average discrepancy is (d) 1.5% for $f = 0.3$, (e) 2.1% for $f = 0.5$, and (f) 5.9% for $f = 0.9$; for scale-free network, the average discrepancy is (g) 2.6% for $f = 0.3$, (h) 4.4% for $f = 0.5$, and (i) 8.2% for $f = 0.9$. Again, the average discrepancy is the least for data-driven network, next lowest for random network and highest for scale-free network. All the average discrepancies are nonetheless low, below 4.5%, except in one case that of the combination of very high value of $f$ and scale-free network. Even for the last combination, the average discrepancy is below 10% and therefore deemed low.

The mismatch that we observe arises because the model equations (refer to Eq (1)) were derived under simplifying assumptions and approximations. The assumption was that the infection status of different nodes constitute independent random variables, but in practice the

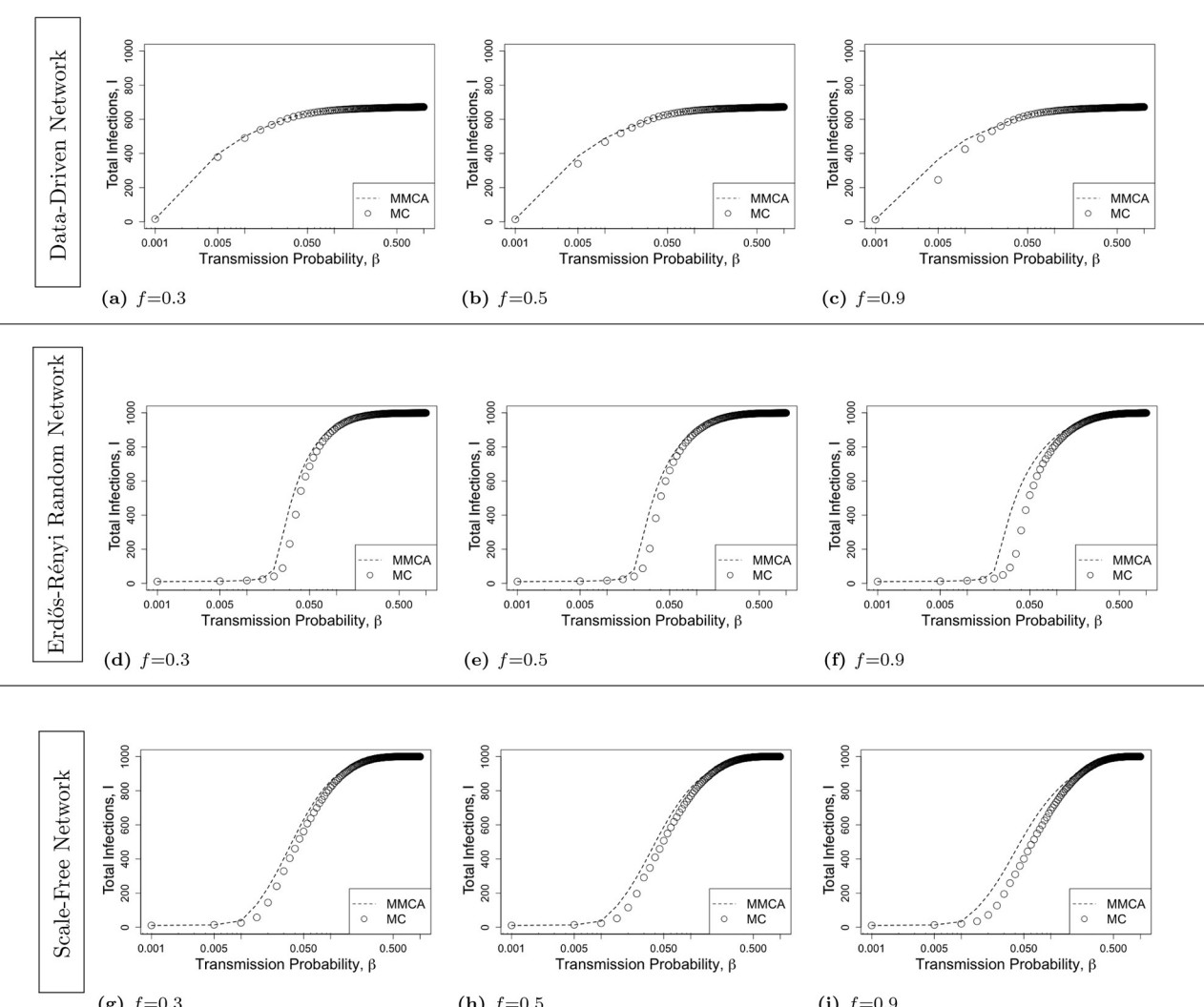

**Fig 3. The total infection count *I* as a function of *β* ∈ [0.001, 1] for *1-hop contact tracing* for various contact networks.** The first, second, and third rows respectively represent Data-Driven network, Erdős-Rényi Random Network, and Scale-Free Network. For each network topology, we consider various values of activation probability *f* = 0.3, 0.5, 0.9. The lines represent the MMCA formulation and the points represent MC simulation. The MMCA formulation closely approximates the MC simulation results under 1-hop contact tracing.

infection statuses of nodes exhibit stochastic correlation and the correlation typically decreases with increase in distance between concerned nodes in the contact graph. This assumption is somewhat commonplace in works that model contact tracing and the spread of epidemic through contacts, and has for example been resorted to in [6]. For *k*-hop contact tracing where *k* > 1, there is an additional approximation: we approximate a cyclic contact graph as an acylic one through a specific construction. The results discussed above show that the discrepancy under 2-hop contact tracing is greater than the one under 1-hop contact tracing, which is likely because of this additional approximation. Furthermore, the results above shows that, for both 1-hop and 2-hop contact tracing, the discrepancy increases as the value of *f* increases. This is because as *f* increases greater number of contacts can be traced, which increases the correlations between the infection status of nodes by facilitating detection of infection in neighbors of detected nodes.

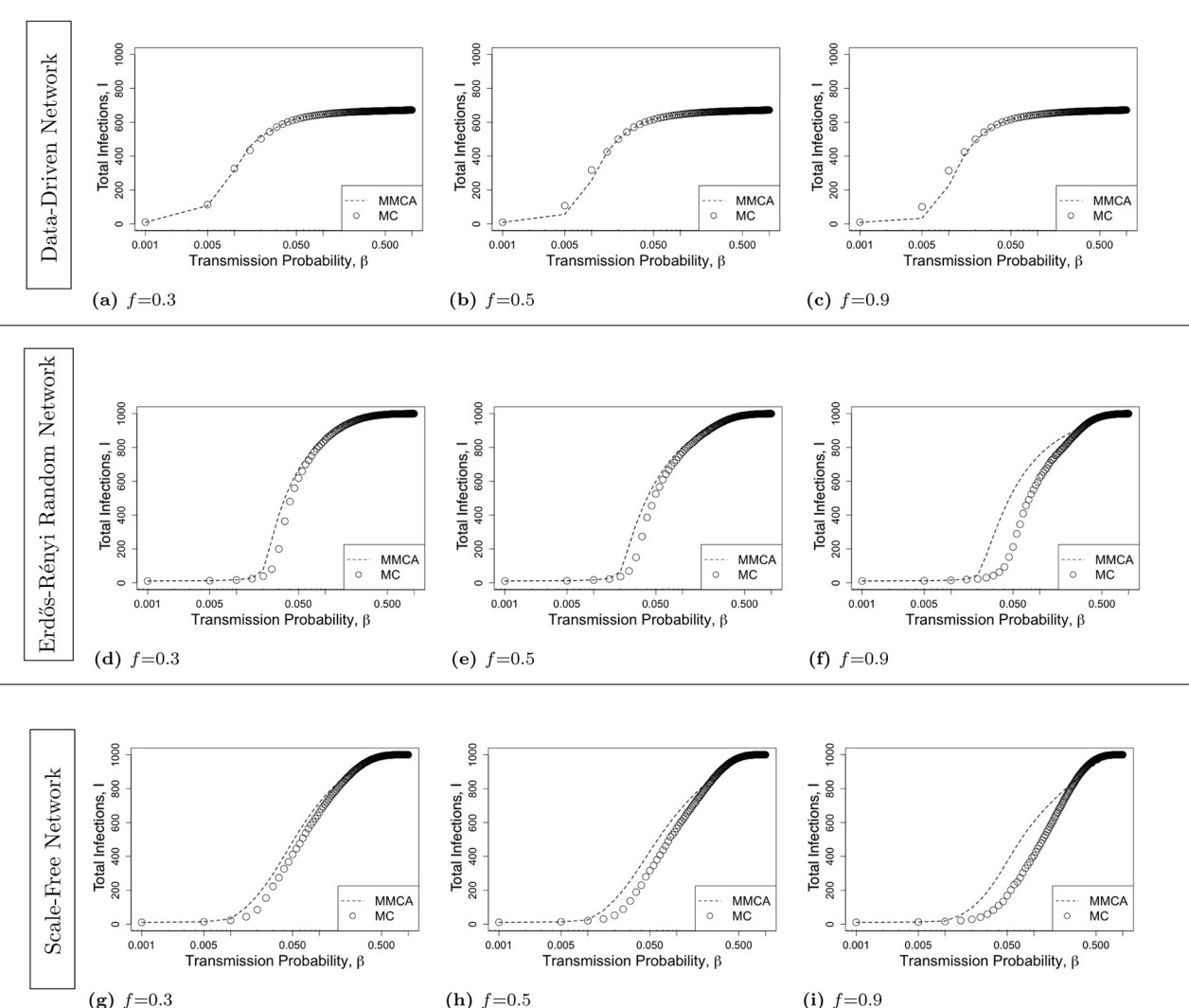

**Fig 4. The total infection count *I* as a function of *β* ∈ [0.001, 1] for *2-hop contact tracing* for various contact networks.** The first, second, and third rows respectively represent Data-Driven network, Erdős-Rényi Random Network, and Scale-Free Network. For each network topology, we consider various values of activation probability *f* = 0.3, 0.5, 0.9. The lines represent the MMCA formulation and the points represent MC simulation. The MMCA formulation closely approximates the MC simulation results under 2-hop contact tracing.

Next, we consider Watts-Strogatz networks with various values of rewiring probability *r*. Fig 5 represents the total infection count as a function of *β* for various values of *r*, 2-hop contact tracing and *f* = 0.3. This shows that average error decreases with increase in *r*; the error is (a) 2.4% for *r* = 0.1, (b) 1.6% for *r* = 0.5, and (c) 1.5% for *r* = 1.0. Fig 6 shows a comparison for 2-hop contact tracing and *f* = 0.7. It also confirms the phenomenon observed in Fig 5; the average error is (a) 5.3% for *r* = 0.1, (b) 4.1% for *r* = 0.5, and (c) 3.5% for *r* = 1.0.

## 3.2 Use of the model for cost-benefit analysis

The Markovian equations enable us to explore the effectiveness of epidemic control via different types of contact tracing strategies on any contact network. Using the data-driven network, we evaluate the impact of different types of contact tracing strategies on benefit (i.e., reduction

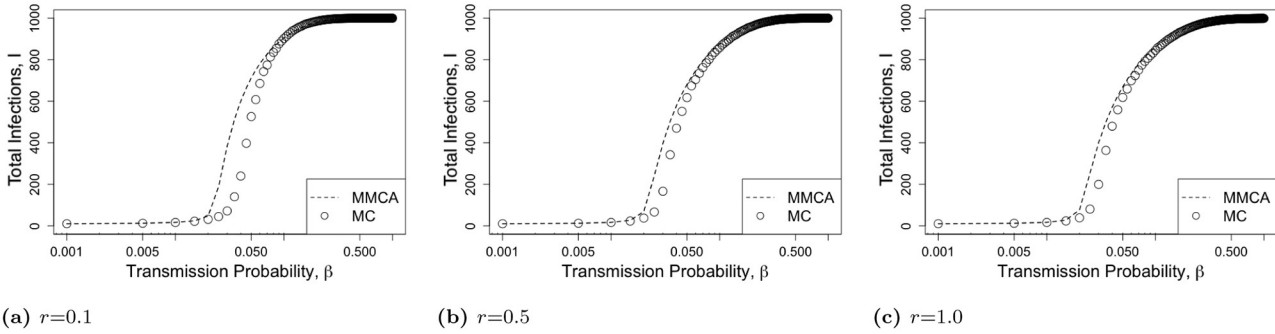

**Fig 5. The total infection count *I* as a function of *β* under 2-hop contact tracing with *f* = 0.3 for various values of rewiring probability (a) *r* = 0.1, (b) *r* = 0.5, and (c) *r* = 1.0 for Watts-Strogatz networks.**

in outbreak size) and costs (i.e., the number of quarantines and tests). In this section, we show that the cost-benefit tradeoff can be enhanced through an implementation of the multi-hop contact tracing. We here assume that a random 1% of the population is initially infected and is in a Latent state.

**3.2.1 Effectiveness of epidemic control via contact tracing.** We first compare the outbreak size (i.e., the total number of infections) over the course of 1 year (i.e., $\tau$ = 365) for different types of contact tracing strategies. We evaluate the ratio $RI_{k\text{-}hop}$ of cases infected via *k*-hop contact tracing by time $\tau$ to the cases infected when no contact tracing is performed, with respect to the activation probability *f* and *attack rate*. $RI_{k\text{-}hop}$ is less (greater, respectively) than 1 if *k*-hop contact tracing results in fewer (greater, respectively) number of infections than 0-hop contact tracing. The attack rate is defined as the proportion of individuals in a population who has been infected during the course of an epidemic by time $\tau$ when no contact tracing is performed, i.e., $I_{0\text{-}hop}/N$, which characterizes the intrinsic speed of virus spread in the absence of contact tracing. We consider *f* in the range of [0.1, 1.0] with uniform interval 0.1, and also consider the attack rate in the range of [0.2, 0.9] by varying *β* with uniform interval 0.0001. Recall that greater the value of *f* greater is the number of contacts available for tracing and more effective contact tracing is expected to be. Fig 7a shows that $RI_{1\text{-}hop}$ is less than 1 in the range of all values of the variables (i.e., *f* and attack rate) we consider, i.e., 1-hop contact tracing enables the reduction in the number of infections across all the ranges. As *f* increases and attack rate decreases, $RI_{1\text{-}hop}$ decreases; in an ideal scenario of the highest *f*, i.e., *f* = 1 and

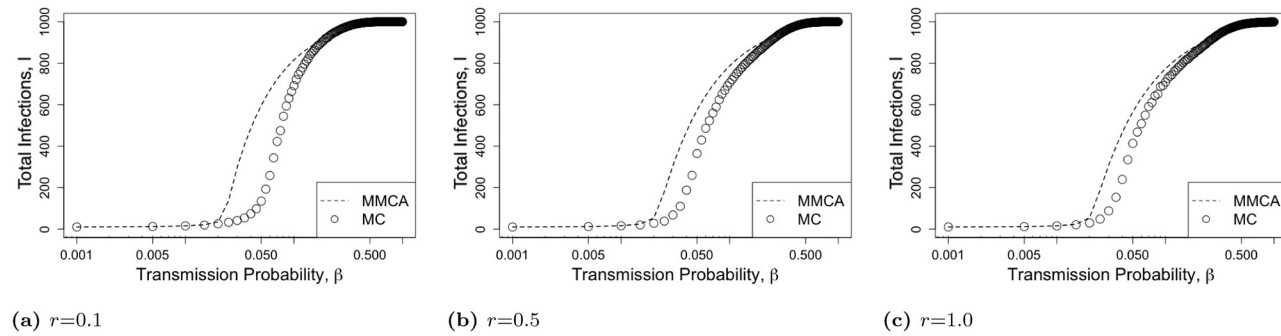

**Fig 6. The total infection count *I* as a function of *β* under 2-hop contact tracing with *f* = 0.7 for various values of rewiring probability (a) *r* = 0.1, (b) *r* = 0.5, and (c) *r* = 1.0 for Watts-Strogatz networks.**

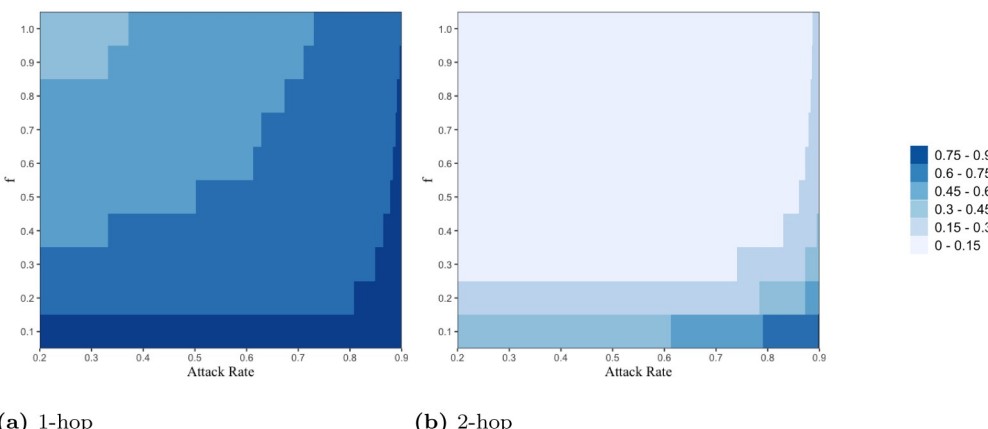

**(a)** 1-hop **(b)** 2-hop

**Fig 7. The ratios (a) $RI_{1\text{-}hop}$ and (b) $RI_{2\text{-}hop}$ are expressed in color as a function of attack rate (x-axis) and activation probability $f$ (y-axis).** We choose different color codes to represent different ranges of values of the ratios (i.e., $RI_{1\text{-}hop}$ and $RI_{2\text{-}hop}$) corresponding to values of $f$ and attack rate. We use the data-driven network for this figure.

the lowest attack rate 0.2, $RI_{1\text{-}hop}$ is lower bounded by 0.41. Fig 7b shows that $RI_{2\text{-}hop}$ is also less than 1 regardless of the values of $f$ and attack rate we consider, and more importantly it shows that $RI_{2\text{-}hop}$ is significantly lower than $RI_{1\text{-}hop}$; $RI_{2\text{-}hop}$ is lower bounded by 0.04. This shows that a slightly more aggressive policy of simply adding one more hop, 2-hop contact tracing, can significantly contain the spread of epidemics. In particular, even in an undesirable case with a high attack rate, a slight increase in $f$ under 2-hop contact tracing scheme can lead to a dramatic decrease in outbreak size.

We now try to understand why 2-hop contact tracing has a significantly lower infection count as compared to 1-hop contact tracing. For this we plot the total number of infections ($I$), the total number of detected infections ($D$), and undetected infections ($U$) over the course of 1 year (i.e., $\tau = 365$) for the two contact tracing strategies (Fig 8). We plot $I$, $D$, $U$ as functions of

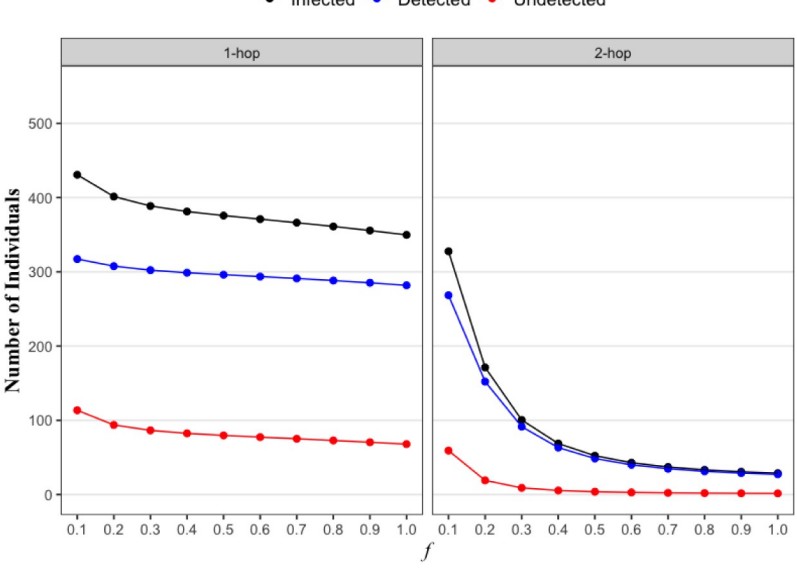

**Fig 8. The expected number of infections (I), and the number of detected infections (D) and undetected infections (U) as a function of activation probability f.** We use the data-driven network for this figure.

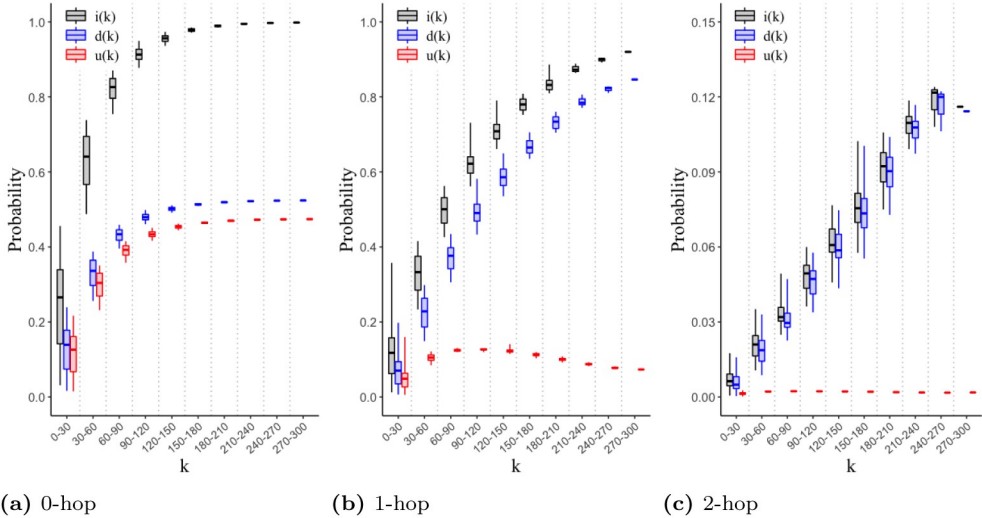

**(a)** 0-hop **(b)** 1-hop **(c)** 2-hop

**Fig 9. Infections, detected infections, and undetected infections as a function of degree $k$ for different contact tracing strategies.** This figure shows the probability that a node of degree $k$ has been infected $i(k)$, detected $d(k)$, and infected but not detected $u(k) := i(k) - d(k)$. We use the data-driven network for this figure.

activation probability $f$ when attack rate is 0.8. The number of infections slowly decreases under 1-hop contact tracing as $f$ increases, but there are still non-negligible number of undetected infections even in an ideal scenario of $f = 1$. The undetected infections continue to spread the infection leading to a large overall infection count. On the other hand, in case of 2-hop contact tracing, even a small increase in $f$ can significantly reduce the number of undetected infections to almost zero. Thus infections are being detected and consequently isolated (quarantined). Thus not many nodes can spread the infection, which consequently significantly reduce the number of infections with even a small increase in $f$. It is clear that 2-hop contact tracing can effectively contain the epidemic even in challenging environments with a small $f$, primarily because it can detect and therefore isolate most of the infected individuals.

Next, we analyze the connectivity pattern of those cases infected and detected for different contact tracing strategies through computing 1) $i(k)$, the probability that a node of degree $k$ has been infected by time $\tau$, 2) $d(k)$, the probability that a node of degree $k$ has been detected by time $\tau$, and 3) $u(k) = i(k) - d(k)$, the probability that a node of degree $k$ has been infected but not detected by time $\tau$. In Fig 9a, we plot $i(k)$, $d(k)$, and $u(k)$ when no contact tracing is performed. We set $f = 0.7$ and the attack rate to 0.8. This shows that $i(k)$ increases with increase in the degree $k$, i.e., the greater number of contacts an individual has, the more likely he is to get infected. We observe that $d(k)$ also increases with increase in degree $k$; this is because nodes with higher number of contacts are more likely to be infected and consequently more likely to be detected after showing symptoms. However, $u(k)$ is as large as $d(k)$. Therefore, a large number of superspreaders (i.e., infected nodes with large degrees) are evading detection when no contact tracing is performed.

An outbreak can be effectively controlled only if the superspreaders are detected early on. When 1-hop contact tracing is performed, $u(k)$ decreases significantly across the values of degree $k$ as compared to the case of no contact tracing (Fig 9b). In particular, $u(k)$ displays a maximum at some degree $k$, and beyond this maximum value, $u(k)$ decreases with increase in $k$. Thus contact tracing is detecting superspreaders of very high degrees. Moreover, as shown in Fig 9c, 2-hop contact tracing demonstrates maximum efficiency. $u(k)$ approaches zero for

all values of degree $k$ meaning that infected nodes have almost zero probability of evading detection.

**3.2.2 Cost-effectiveness of contact tracing.** We have shown that detecting and isolating the infected individuals through 2-hop contact tracing can stop greater number of infected individuals from further spreading the disease as compared to 1-hop contact tracing. But 2-hop contact tracing can also involve an increase in the number of people tested and quarantined as compared to 1-hop and no contact tracing schemes. We show how our framework can determine if this is indeed the case by allowing the evaluation of the above quantities which represent the costs associated with contact tracing strategies. This in turn helps public health authorities decide the level of contact tracing a public health system ought to opt for based on the magnitude of the costs it can afford. Using the results from our framework, we argue that 2-hop provides a more favorable cost-benefit tradeoff compared to 0-hop and 1-hop.

We first compare the quarantining cost, i.e., the total number of people that need to be quarantined (equivalently, the total number of people detected), over the course of 1 year (i.e., $\tau = 365$) for different types of contact tracing strategies. We compute the ratio $RD_{k\text{-}hop}$ of the total number of individuals detected by time $\tau$ under $k$-hop contact tracing to that under 0-hop.

We first plot $RD_{1\text{-}hop}$ with respect to $f$ and attack rate. Fig 10a shows that as $f$ decreases and attack rate increases, $RD_{1\text{-}hop}$ increases; $RD_{1\text{-}hop}$ is greater than 1 in higher region of attack rate and lower region of $f$, i.e., 1-hop contact tracing quarantines a greater number of individuals compared to the case of no contact tracing. In particular, when the attack rate is very high, 1-hop contact tracing quarantines a greater number of individuals regardless of the value of $f$. On the other hand, Fig 10b shows that $RD_{2\text{-}hop}$ is less than 1, and that $RD_{2\text{-}hop}$ is significantly lower than $RD_{1\text{-}hop}$, across almost all values of $f$ and attack rate we consider. Thus, 2-hop contact tracing substantially decreases the overall number of quarantines (i.e. quarantine cost) compared to traditional 1-hop contact tracing regardless of $f$ and attack rate, despite the fact that 2-hop controls the outbreak more effectively. Moreover, 2-hop contact tracing significantly reduces the number of quarantines required even compared to the case of no contact

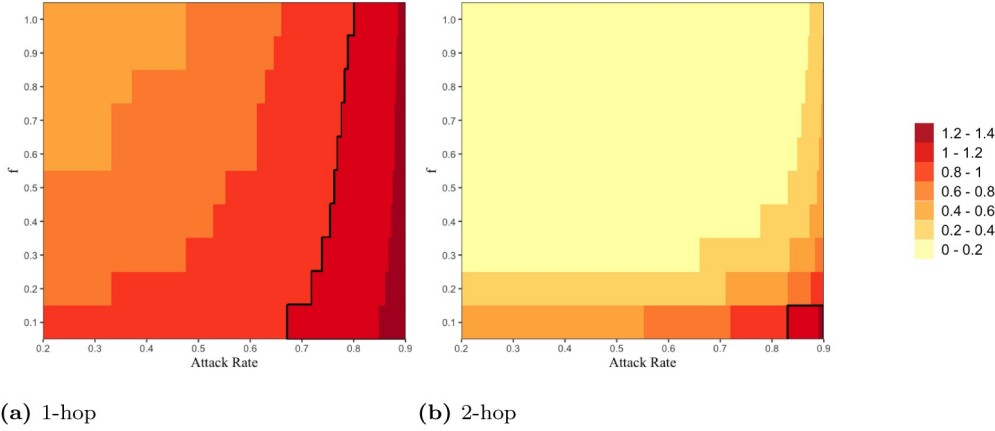

**(a)** 1-hop **(b)** 2-hop

**Fig 10. The ratios (a) $RD_{1\text{-}hop}$ and (b) $RD_{2\text{-}hop}$ are expressed in color as a function of attack rate (x-axis) and activation probability $f$ (y-axis).** We choose different color codes to represent different ranges of values of the ratios (i.e., $RD_{1\text{-}hop}$ and $RD_{2\text{-}hop}$) corresponding to values of $f$ and attack rate. We have highlighted the case $RD_{k\text{-}hop} = 1$ (black line) signaling that 0-hop and $k$-hop quarantine the same number of individuals. We use the data-driven network for this figure.

tracing in all but extremely challenging environments; see bottom right region surrounded by black line in Fig 10b which corresponds to very low $f$ (therefore most contacts can not be traced) and very high attack rate.

We now explain the apparently counter-intuitive result, that is, as to why 2-hop contact tracing quarantines substantially fewer number of individuals as compared to 1-hop or 0-hop contact tracing. Recall that $RD_{k-hop} = RD_{k-hop}^{CT} + RD_{k-hop}^{S}$, where $RD_{k-hop}^{CT}$ is the ratio of the number of individuals detected via $k$-hop contact tracing by time $\tau$ to the number of individuals detected under 0-hop and $RD_{k-hop}^{S}$ is the ratio of the number of individuals detected because they tested as a result of showing symptoms by time $\tau$ to the number of individuals detected under 0-hop. We plot $RD_{k\text{-}hop}$, $RD_{k-hop}^{S}$, and $RD_{k-hop}^{CT}$ for different contact tracing strategies, as a function of $f$ when attack rate is 0.8 (Fig 11). For 1-hop contact tracing (left plot in Fig 11), the number of quarantines required slowly decreases (from 1.13 to 1) as $f$ increases. In particular, $RD_{1-hop}^{CT}$ increases with $f$ while $RD_{1-hop}^{S}$ decreases. Nevertheless, both detection via contact tracing and symptom-based detection contribute significantly to the number of quarantines for all values of $f$. On the other hand, in case of 2-hop contact tracing (right plot in Fig 11), even a small increase in $f$ reduces $RD_{2-hop}^{S}$ to a value close to 0. This suggests that the number of individuals with symptoms (and therefore the number of individuals infected as a certain fixed fraction of infected individuals develop symptoms) reduce to near 0 with even a modest increase in $f$ (the probability that a contact can be traced). Thus, the overall infection count becomes very small with a modest increase in $f$. Once the overall infection count becomes small, the number of individuals detected would have to become small, this in turn reduces the number of primary contacts and therefore the number of secondary contacts traced. Thus, $RD_{2-hop}^{CT}$ also rapidly decreases with increase in $f$. Naturally then $RD_{2\text{-}hop}$, which equals $RD_{2-hop}^{CT} + RD_{2-hop}^{S}$, also significantly reduces as $f$ increases.

It however turns out that 2-hop contact tracing needs to test a greater number of individuals compared to 1-hop contact tracing. Fig 12 shows that the ratio of total tests via 2-hop contact tracing to total tests via 1-hop (i.e., $T_{2\text{-}hop}/T_{1\text{-}hop}$) is greater than 1 across the values of $f$ and

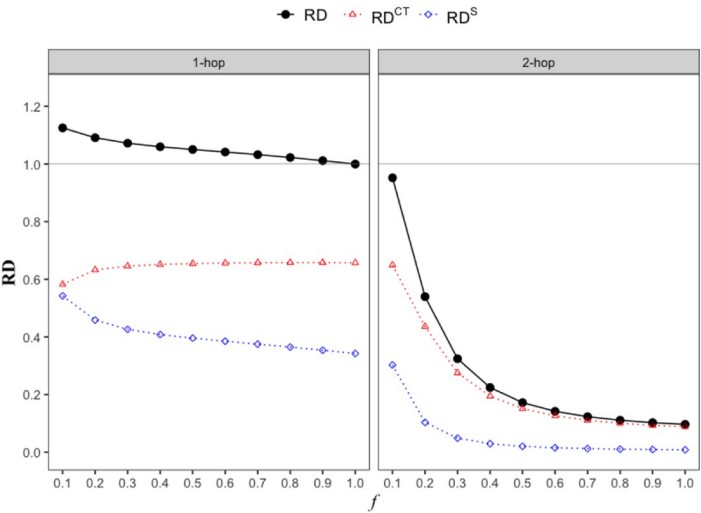

**Fig 11. $RD_{k\text{-}hop}$, $RD_{k-hop}^{CT}$, $RD_{k-hop}^{S}$ with respect to the activation probability $f$.** We use the data-driven network for this figure.

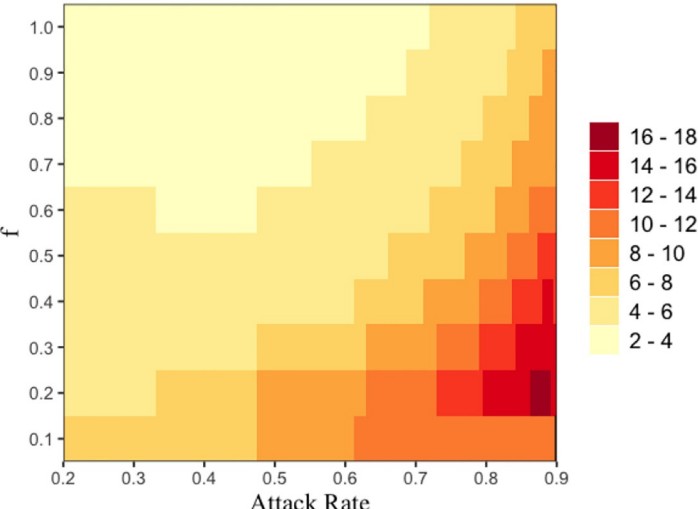

**Fig 12. The ratio of total tests via 2-hop contact tracing to total tests via 1-hop (i.e., $T_{2\text{-}hop}/T_{1\text{-}hop}$) are expressed in color as a function of attack rate (x-axis) and activation probability $f$ (y-axis).** We choose different color codes to represent different ranges of values of the ratio (i.e., $T_{2\text{-}hop}/T_{1\text{-}hop}$) corresponding to values of $f$ and attack rate. We use the data-driven network for this figure.

attack rate we consider. In addition, the ratio decreases with $f$ and increases with attack rate. Thus, 2-hop contact tracing incurs greater testing cost as compared to 1-hop contact tracing.

We now argue that for any contact tracing scheme, the number of individuals quarantined should be considered a more significant component of the overall cost than the number of tests. Note that various countries, at least the developed ones, could quickly scale up testing capacity within a few months of start of Covid-19. In general, an individual finds it more disruptive to be quarantined than to be tested because the former usually involves several days of not being able to discharge regular professional and social responsibilities. The following have been documented in 2 years of Covid. Long-term quarantines significantly increase non-COVID-19 fatalities [17]. During lockdowns, which represents long-term quarantines, there has been an increase in drug overdose deaths (eg, in Ontario and British Columbia), possibly as a result of isolation [18]. Lockdowns have been associated with an increase in domestic violence (e.g., 30% more in France and 25% more in Argentina) [19]. Lockdowns have wreaked havoc on economies around the world, especially in developing nations [20, 21]. Quantifying the negative effects of all of the aforementioned are challenging. Overall, nevertheless, the substantially greater quarantine cost may be more important than other costs.

Thus, since 2-hop contact tracing significantly lowers 1) the most significant component of the costs, and 2) the overall infection count compared to 1-hop and 0-hop, the cost-benefit tradeoff for 2-hop may be considered more favorable compared to 1-hop and 0-hop.

### 3.3 Computation time

We finally compare the computational time required in generating the output of the MMCA model formulation with that of running 500 runs of MC simulations. We observe the change in computation time required for both with the size of Erdős Rényi random network, from 1,000 to 20,000 nodes and fixed mean degree $\langle k \rangle = 8$. We set $\beta = 0.001$ and $f = 0.5$. Fig 13 shows that computation times for both MC simulations and MMCA formulation increase linearly with the network size (i.e., the numbers of nodes), but the computation time for MC

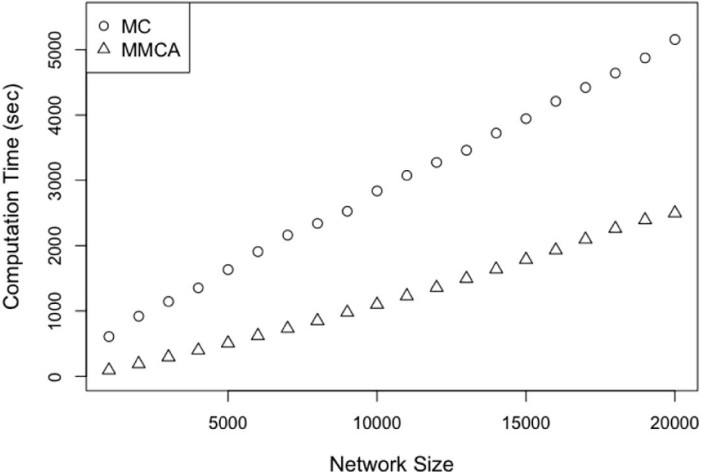

**Fig 13. Computation time as a function of population size for Erdős Rényi random network.**

simulations increases much more rapidly. In addition, MC simulations usually requires a significantly larger number of iterative runs for accurate statistics based on samples (simulations run). As a result, the computation time for MC simulation increases even more steeply with network size if a greater number of runs are made, and the difference in computation time between MC simulation and MMCA may become greater as the former requires greater accuracy. In practice, contact networks can be large, e.g., population in large cities in US are of the order of millions (e.g., New York city had more than 8 million residents in 2021) while small towns in US have thousands of residents. Thus, MMCA can scale to contact networks that arise in practice in a computationally efficient manner, which MC simulations may not be able to accommodate in reasonable computation times. We used Amazon Web Services (AWS) m5a.large instances to compare the computation time.

## 4 Comparing contact tracing strategies when number of tests are equal and under testing fatigue

We have shown in Section 3.1 that 2-hop contact tracing substantially reduces the outbreak size and quarantine cost as compared to 1-hop and 0-hop contact tracing, but incurs a greater number of tests overall. We now examine if the above benefits of 2-hop contact tracing are only or primarily because it tests a larger number of individuals or because it intelligently selects who to test. Towards that end, we first generalize the testing strategies to consider a combination of contact tracing and additional random testing individuals and design a numerical framework to evaluate the combination (Section 4.1). This generalization allows us to approximately equalize the overall number of tests of 1) 2-hop contact tracing and 2) the combination of 1-hop contact tracing and additional random testing by choosing the number of additional random tests in the latter. Once the overall number of tests has been equalized we compute and compare the outbreak size under each; the comparison reveals that 2-hop contact tracing still substantially reduces the outbreak size as compared to the combination, which in turn suggests that 2-hop contact tracing is able to significantly reduce the outbreak size because it judiciously deploys its additional tests as compared to 1-hop contact tracing. We subsequently obtain a realistic generalization wherein once an individual tests negative he does not test further for a certain duration unless he develops symptoms in the interim period;

the size of the duration can be appropriately selected to satisfy budgets on average daily overall test count (Section 4.2). Our numerical computation reveals that 2-hop contact tracing continues to considerably reduce the outbreak size as compared to 1-hop contact tracing even when it satisfies stringent budgets on average daily overall test count and even as its number of tests becomes close to that for 1-hop contact tracing. This again shows that 2-hop contact tracing attains better cost-benefit tradeoff than 1-hop contact tracing because it judiciously selects which nodes to test. Both these generalizations demonstrate that our numerical computation framework is flexible enough to accommodate several features of reality.

## 4.1 Combination of contact tracing and random testing

We consider a combination of contact tracing and random testing in which a node can be tested with probability $\delta$ for each time step even if he is not in a $k$-hop neighborhood of a detected node or even if he does not show symptoms. We consider $k = 1, 2$ in this section, but the framework directly generalizes to larger values of $k$ through an extension of the $k$-hop contact tracing framework provided in the S1 Text. Specifically, earlier, pre-symptomatic and asypmtomatic nodes could be detected only through contact testing; now such nodes can be detected either through random testing or through contact tracing. Earlier symptomatic nodes could be detected through contact tracing, or testing with probability $\omega$ because of showing symptoms; now they may also be detected through random testing.

Under this approach, only the value of the probability that a node $i$ in the state $X$ is detected through testing at time $t$, $\Pi_i^{X \to D}(t)$, need to be adapted. For $k = 1$,

$$\Pi_i^{X \to D}(t) = \begin{cases} 1 - (1-\delta) \prod_{j=1}^{N} [1 - A_{ij} f \rho_j^D(t)], & X \in \{I_p, I_a\} \\ 1 - (1-\delta)(1-\omega) \prod_{j=1}^{N} [1 - A_{ij} f \rho_j^D(t)], & X = I_s. \end{cases} \qquad (16)$$

When $k = 2$, the term in bracket $[\ldots]$ need to be replaced by $\prod_{j \in \overline{V}_i^{(1)} \setminus \overline{V}_i^{(0)}}$ in Eq 4. The rest of the framework presented in Section 2.1 holds as is once the modified value of $\Pi_i^{X \to D}(t)$ is used therein.

We use this framework to compare 2-hop contact tracing with the combination of 1-hop contact tracing and random testing. For the latter, i.e., '1-hop + random testing' approach, we choose $\delta$ so that the total number of tests for '2-hop' and '1-hop + random testing' are almost the same (i.e., $\frac{T_{2-hop}}{T_{1-hop+\text{random}}} \simeq 1$) (Note that $T_{1-hop+ \text{ random}}$ can be obtained from Section 2.1 by using $\Pi_i^{X \to D}(t)$ computed above for $k = 1$ in place of $\Pi_i^{X \to D}(t)$ computed for $k = 1$ in Section 2.1.). We choose $\tau$ as 365 days as before. We then compare the total number of infections for '2-hop' and '1-hop + random testing'. Fig 14 shows that, despite both 2-hop' and '1-hop + random testing' incurring the same number of tests, 2-hop contact tracing provides significant advantage over '1-hop + random testing' strategy in terms of outbreak size for all attack rates. This demonstrates that the cost-benefit tradeoff can be enhanced through an implementation of the multi-hop contact tracing.

## 4.2 Contact tracing under testing fatigue

We here study a realistic generalization of our strategies which reduces the average daily number of tests of contact tracing strategies and thereby satisfy budgetary constraints on this average through appropriate selection of parameters. In Section 2, for the sake of model simplicity, we had assumed that individuals can be repeatedly tested if traced from contacts who test positive regardless of the outcome of their previous tests. But in reality, individuals develop a testing fatigue if they are frequently subjected to repeated tests, and the tests are

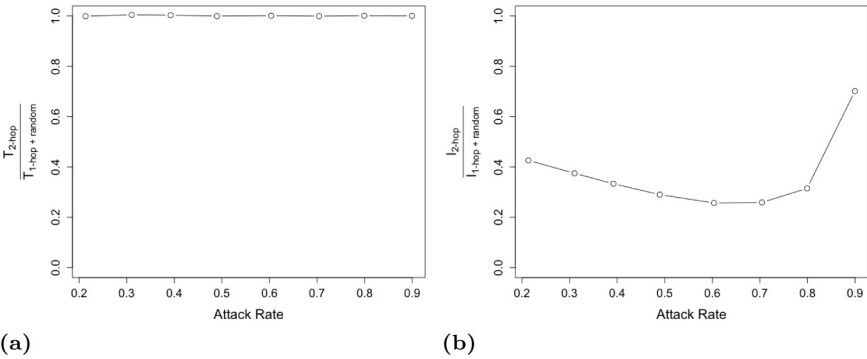

**Fig 14.** (a) The ratio of number of tests via 2-hop contact tracing to that under the combination of 1-hop contact tracing and random testing, $\frac{T_{2-hop}}{T_{1-hop+\text{random}}}$. (b) The ratio of cases infected via 2-hop contact tracing to the cases infected via '1-hop + random testing', $\frac{I_{2-hop}}{I_{1-hop+\text{random}}}$. We use the data-driven network for this figure.

negative. Due to this fatigue, individuals may therefore refuse to be tested shortly after a negative test, unless they develop symptoms, even if traced from contacts who test positive. Public health bodies may also avoid recommending tests for an individual shortly after he tests negative unless he develops symptoms, to avoid testing fatigue, elicit his cooperation in future, and satisfy constraints on the overall number of tests. We therefore consider that an individual is not tested for a random duration after he is tested negative unless he develops symptoms, even if he is traced from a 1-hop or 2-hop contact; if he tests positive he is quarantined and not tested again as we don't consider reinfection. The random duration is geometrically distributed with mean $1/\xi$. We show that $\xi$ can be tuned to have both 1-hop and 2-hop satisfy budgetary constraints on daily average number of tests, reduce the difference between the daily average number of tests of 2-hop and 1-hop, and still reduce the outbreak size through 2-hop compared to 1-hop.

Modeling the above requires a generalization of our state diagram and modeling equations (Fig 15). We consider that right after an individual tests negative he becomes "not-ready-to-test" for a random time. In this condition he does not test unless he develops symptoms. We split the states up to symptomatic (namely, susceptible, latent, presymptomatic, asymptomatic) into two distinct states representing whether 1) individuals are ready to test or 2) otherwise. Thus, susceptible state is split into ready-to-test susceptible ($S1$) and not-ready-to-test susceptible ($S2$), latent state is split into ready-to-test latent ($L1$) and not-ready-to-test latent ($L2$), etc.

Once an $S1$ ($S2$, respectively) individual is infected, he transitions to $L1$ ($L2$, respectively). An $S1$ ($L1$, respectively) individual can be tested as a result of contact tracing, if he is tested, he tests negative and transitions to $S2$ ($L2$, respectively). An $S2$ ($L2$, respectively) individual is not tested, even if he is traced from a 1-hop or 2-hop contact, but he transitions to $S1$ ($L1$, respectively) at any given time with probability $\xi$. Basically $S1$'s possible next states are $S2$, $L1$, $S2$'s possible next states are $S1$, $L2$.

Presymptomatics (Asymptomatics, respectively) are also split into ready to test and not ready to test versions: $I_p1$, $I_p2$ ($I_a1$, $I_a2$, respectively). Upon progression of the disease, $L1$ transitions to $I_p1$ or $I_a1$, $L2$ transitions to $I_p2$ or $I_a2$. An individual in $I_p1$, $I_a1$ state is tested if he is traced from a 1-hop or 2-hop contact, when he is tested he tests positive and transitions to the detected state. An individual in $I_p2$ ($I_a2$, respectively) state is not tested even if he is traced from a 1-hop or 2-hop contact, but at each time $t$ he becomes ready to test, i.e., transitions to

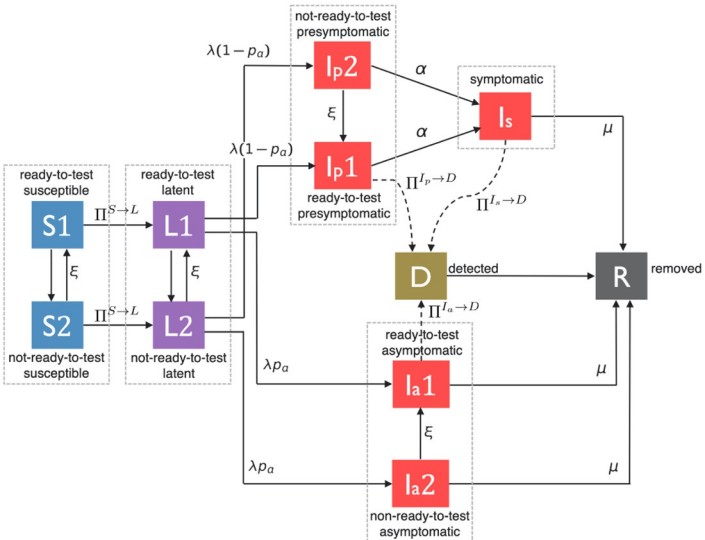

**Fig 15. A pictorial illustration of state transitions when there is test fatigue.**

$I_p1$ ($I_a1$, respectively), with probability $\xi$. A symptomatic ($I_s$) is always ready to test as he experiences symptoms. Thus, we don't split the symptomatic state, and upon progression of the disease, both $I_p1$, $I_p2$ become symptomatic. We don't split the removed ($R$) state either. Thus, $I_p1$'s next states are $D$, $I_s$, $I_p2$'s next states are $I_p1$, $I_s$, $I_a1$'s next states are $D$, $R$, $I_a2$'s next states are $I_a1$, $R$.

The probability that a susceptible node $i$ is infected (and then transits to Latent state) at time $t$ is the same for $S1$, $S2$, as the contact rates and the probability that a contact transmits the contagion do not depend on the readiness to be tested. Hence we still denote it as $\Pi_i^{S \to L}(t)$, i.e., we use a generic $S$ instead of $S1$, $S2$. But its value will be different from when the states are not partitioned, i.e., when everyone is willing to be tested all the time. When states are partitioned, its value is:

$$\Pi_i^{S \to L}(t) = 1 - \prod_{j=1}^{N}[1 - A_{ij}\beta\{\rho_j^{I_p1}(t) + \rho_j^{I_p2}(t) + \rho_j^{I_a1}(t) + \rho_j^{I_a2}(t) + \rho_j^{I_s}(t)\}], \tag{17}$$

where as before $\rho_i^X(t)$ is the probability that node $i$ is in state $X$ at time $t$, $A = (A_{jk})$ is the adjacency matrix of the original graphs $G(V, E)$, in which $|V| = N$, and $A_{ij} = 1$ if there exists an edge between node $i$ and node $j$, and $A_{ij} = 0$ otherwise.

Recall also that $\Pi_i^{X \to D}(t)$ and $\Pi_i^{X \to T}(t)$ are the probabilities that a node $i$ in the state $X$ is detected and tested at time $t$, respectively. Note that state names containing the number 2 are neither tested nor detected. To simplify the notation, numbers corresponding to partitions of states (i.e., 1 and 2) are excluded from the notation of these probabilities (e.g., $\Pi_i^{I_p \to D}(t)$ instead of $\Pi_i^{I_p1 \to D}(t)$). The expressions for these probabilities, $\Pi_i^{X \to D}(t)$ and $\Pi_i^{X \to T}(t)$, are the same as expressions as in equations in Section 2. The dynamics of spreading of epidemics and different

types of contact tracing intervention can be adapted as follows:

$$\rho_i^{S1}(t+1) = [1 - \Pi_i^{S \to L}(t)][1 - \Pi_i^{S \to T}(t)]\rho_i^{S1}(t) + [1 - \Pi_i^{S \to L}(t)]\xi\rho_i^{S2}(t)$$

$$\rho_i^{S2}(t+1) = [1 - \Pi_i^{S \to L}(t)](1 - \xi)\rho_i^{S2}(t) + [1 - \Pi_i^{S \to L}(t)]\Pi_i^{S \to T}(t)\rho_i^{S1}(t)$$

$$\rho_i^{L1}(t+1) = (1 - \lambda)[1 - \Pi_i^{L \to T}(t)]\rho_i^{L1}(t) + \Pi_i^{S \to L}(t)\rho_i^{S1}(t) + (1 - \lambda)\xi\rho_i^{L2}(t)$$

$$\rho_i^{L2}(t+1) = (1 - \lambda)(1 - \xi)\rho_i^{L2}(t) + \Pi_i^{S \to L}(t)\rho_i^{S2}(t) + (1 - \lambda)\Pi_i^{L \to T}(t)\rho_i^{L1}(t)$$

$$\rho_i^{I_p1}(t+1) = [1 - \Pi_i^{I_p \to D}(t)](1 - \alpha)\rho_i^{I_p1}(t) + \lambda(1 - p_a)\rho_i^{L1}(t) + \xi(1 - \alpha)\rho_i^{I_p2}(t)$$

$$\rho_i^{I_p2}(t+1) = (1 - \xi)(1 - \alpha)\rho_i^{I_p2}(t) + \lambda(1 - p_a)\rho_i^{L2}(t) \qquad (18)$$

$$\rho_i^{I_a1}(t+1) = [1 - \Pi_i^{I_a \to D}(t)](1 - \mu)\rho_i^{I_a1}(t) + \lambda p_a\rho_i^{L1}(t) + \xi(1 - \mu)\rho_i^{I_a2}(t)$$

$$\rho_i^{I_a2}(t+1) = (1 - \xi)(1 - \mu)\rho_i^{I_a2}(t) + \lambda p_a\rho_i^{L2}(t)$$

$$\rho_i^{I_s}(t+1) = [1 - \Pi_i^{I_s \to D}(t)](1 - \mu)\rho_i^{I_s}(t) + [1 - \Pi_i^{I_p \to D}(t)]\alpha\rho_i^{I_p1}(t) + \alpha\rho_i^{I_p2}(t)$$

$$\rho_i^D(t+1) = \Pi_i^{I_s \to D}(t)\rho_i^{I_s}(t) + \Pi_i^{I_a \to D}(t)\rho_i^{I_a1}(t) + \Pi_i^{I_p \to D}(t)\rho_i^{I_p1}(t)$$

$$\rho_i^R(t+1) = \rho_i^R(t) + \rho_i^D(t) + \mu[1 - \Pi_i^{I_s \to D}(t)]\rho_i^{I_s}(t) + \mu[1 - \Pi_i^{I_a \to D}(t)]\rho_i^{I_a1}(t) + \mu\rho_i^{I_a2}(t).$$

We consider an example attack rate, 0.21. We first assume that an individual who tests negative can be tested again after 14 days on average ($\xi = 1/14$), which is reasonable. This corresponds to a situation where average number of tests per day is less than 27 for 2-hop and 1-hop (the left in Fig 16a). Considering that the population size is 672, the average daily testing load is less than 4% of the total population. However, as shown on the right in Fig 16a, 2-hop can reduce the outbreak size by up to 73% compared to 1-hop and 87% compared to 0-hop.

We note that as $\xi$ decreases average number of tests per day decreases for both 1-hop and 2-hop, and the average number of tests for both become closer and closer, but throughout 2-hop considerably reduces the outbreak size as compared to 1-hop. When $\xi = 1/60$, the average number of tests per day is less than 15 for both 1-hop and 2-hop; 2-hop can reduce the

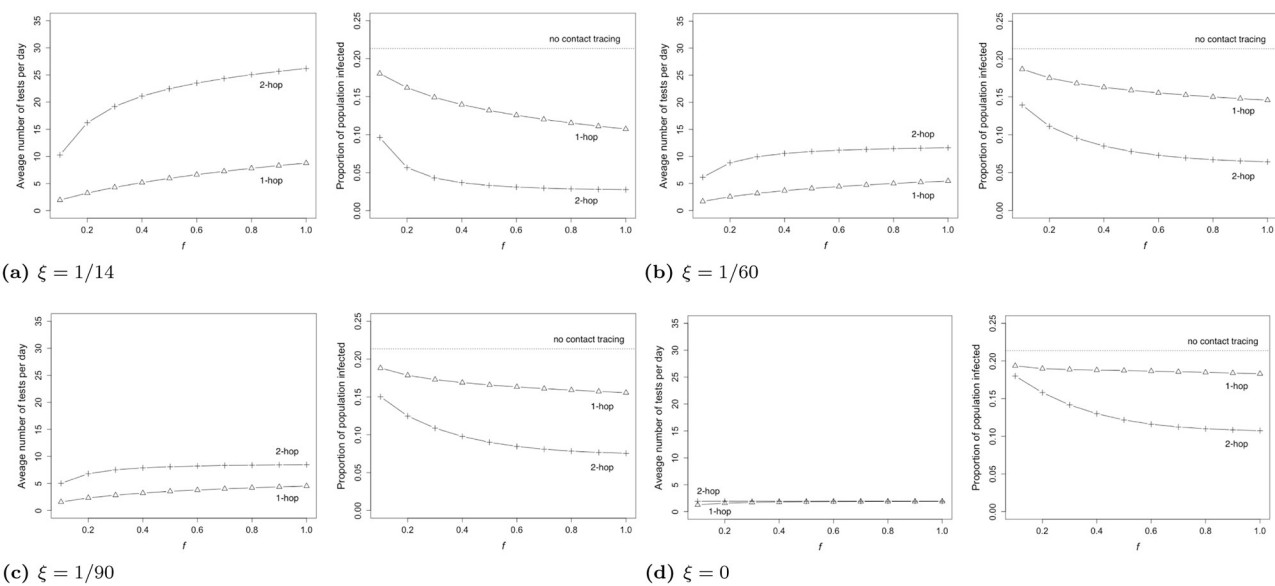

**(a)** $\xi = 1/14$

**(b)** $\xi = 1/60$

**(c)** $\xi = 1/90$

**(d)** $\xi = 0$

**Fig 16. The average number of tests per day and fraction of the populace infected via 1-hop and 2-hop as a function of activation probability *f*.**
Figures (a), (b), (c), and (d) show the results for $\xi = 1/14$, $\xi = 1/60$, $\xi = 1/90$ and $\xi = 0$, respectively. The horizontal dotted lines in the figures on the right represent the number of individuals infected when no contact tracing is performed. We use the data-driven network for this figure.

outbreak size by up to 55% compared to 1-hop and 69% compared to 0-hop (the right in Fig 16b). When $\xi = 1/90$, the average number of tests per day is less than 10; 2-hop can reduce the outbreak size by up to 51% compared to 1-hop and 64% compared to 0-hop (the right in Fig 16c). As an example of extreme case, to further constrain the testing budget, we assume that $\xi = 0$, i.e., individuals who have been tested once will not be tested again. This corresponds to a situation where the average number of tests per day is less than 2, and the difference in the number of tests between 1-hop and 2-hop becomes insignificant (the left in Fig 16d). Nevertheless, the right in Fig 16d shows that 2-hop can reduce the outbreak size by up to 41% compared to 1-hop and 49% compared to 0-hop.

In conclusion, one can satisfy desired constraints on the average number of tests per day by appropriately choosing $\xi$, and still attain a considerable reduction in outbreak size through 2-hop contact tracing over 1-hop contact tracing.

## 5 Conclusion and discussion

In this study, we provide a mathematical framework that computes key attributes for multi-hop contact tracing, by combining the multi-hop contact tracing dynamics and the virus transmission mechanism using microscopic Markov chain approach (MMCA). We first consider 2-hop contact tracing and subsequently generalize it to $k$-hop contact tracing for completeness of the formulation. We utilize our formulation to compare 2-hop contact tracing with 1-hop and 0-hop, and show that 2-hop contact tracing significantly enhances cost-benefit tradeoff as compared to traditional 1-hop contact tracing. Considering a human contact network generated from real-world data, we show that 2-hop contact tracing can reduce the number of infections by more than 80% while reducing quarantine costs by more than 80% compared to case of no contact tracing in large ranges of parameters. This dramatic enhancement of cost-benefit tradeoff accomplished by 2-hop contact tracing alone suggests that contact tracing with larger number of hops would be redundant for the contact network we considered.

We use the same contact network and 2-hop contact tracing scheme to shed light on the mechanisms behind the effectiveness of multi-hop contact tracing. We show that, under 1-hop contact tracing, the number of infections declines, but slowly, as contact information that can be identified by health authorities increases. Even when all contact information can be identified by a health authority, there are still a sizable proportion of undetected infections. In contrast, for 2-hop contact tracing, even a modest increase in the identifiable contact information can drastically lower the number of undetected infections to almost zero and therefore significantly reduce the number of infections. Furthermore, we show that superspreaders (i.e., infected nodes with large degrees) have almost zero probability of evading detection in 2-hop contact tracing while they can evade detection with non-negligible probabilities in 1-hop contact tracing. Thus, in comparison to 1-hop, 2-hop is significantly more effective in controlling an outbreak. Despite the better efficiency of 2-hop, we show that 2-hop contact tracing quarantines substantially fewer number of individuals as compared to 1-hop. Since the overall infection count becomes very small with a modest increase in contact information that can be identified by health authorities, the number of individuals detected would have to become small, this in turn reduces the number of primary contacts and therefore the number of secondary contacts traced, and therefore the overall number of individuals quarantined. Overall, the cost-benefit tradeoff for 2-hop can be considered significantly more favorable compared to 1-hop.

We now describe some limitations of our work which in turn identify directions for future research. We have implicitly assumed that any number of contacts can be traced on any given day. But public health systems may have constraints on the number of contacts

that can be traced per day owing to limitation on tracing personnel for example. There may be constraints on the average number of contacts that can be traced per day which can be satisfied through an appropriate choice of *f*. As *f* decreases, average number of contacts traced per day decreases. There may also be hard upper bound on the number of contacts that can be traced on each day. In this case, contact tracing strategies need to be adapted to satisfy such constraints. This constitutes a direction for future research. Such constraints however become less stringent if contacts are traced through digital tools, e.g, digital contact tracing apps. For example, digital tools were utilized in Vietnam to execute multi-hop contact tracing [5]. But then not all contacts can be digitally traced either as not everyone downloads such apps and the willingness is different in different countries. In Singapore, for example, more than 92% of the population aged six and above had downloaded the government's contact tracing app on their smartphones [22], but the percentage has been lower in other countries. The nature of the constraints depend on the degree of reliance of the public health system on manual tracing.

When evaluating the effectiveness of contact tracing in Section 3, we had assumed that $\beta_{I_s} = \beta_{I_a} = \beta_{I_p} = \beta$ by following the parameter setup in study [7], although we had allowed these parameters to be different while formulating our model in Section 2 In reality, different types of contacts may pass on infection with different probabilities. This transmission probability may also depend on a range of factors, such as whether the individuals observe social distancing and wear protective equipment, and varies from one venue to another. Explicitly investigating the impact of non-uniform transmission probabilities constitute directions for future research.

We have considered two public health cost metrics, quarantine cost and testing cost. There are several other cost metrics that are of interest from a public health perspective such as overall hospitalization and death counts. Hospitalizations are of interest particularly because public healthcare systems have limited hospitalization capacity. Hospitalization and death counts can be investigated by adding new states to the model e.g., a hospitalization state can be added between symptomatic and removed states, the hospitalization state can lead into dead and recovered states, etc. Comparing different contact tracing strategies from the point of view of those costs constitutes a direction of future research.

Our computations show that the discrepancy between our MMCA formulation and MC simulations under 2-hop contact tracing is somewhat greater than that under 1-hop contact tracing. This is because *k*-hop contact tracing involves an acyclic graph approximation for $k > 1$, which it does not for $k = 1$. The magnitude of the discrepancy may further increase as the number of hops increases. This constitutes a limitation of our approach, which future research may be able to surmount.

## Supporting information

**S1 Text. Probability of detection for *k*-hop contact tracing.**
(PDF)

## Author Contributions

**Conceptualization:** Jungyeol Kim, Shirin Saeedi Bidokhti, Saswati Sarkar.

**Formal analysis:** Jungyeol Kim.

**Funding acquisition:** Shirin Saeedi Bidokhti, Saswati Sarkar.

**Methodology:** Jungyeol Kim, Shirin Saeedi Bidokhti, Saswati Sarkar.

**Software:** Jungyeol Kim.

**Supervision:** Shirin Saeedi Bidokhti, Saswati Sarkar.

**Validation:** Shirin Saeedi Bidokhti, Saswati Sarkar.

**Visualization:** Jungyeol Kim.

**Writing – original draft:** Jungyeol Kim.

**Writing – review & editing:** Shirin Saeedi Bidokhti, Saswati Sarkar.

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
