## [Decision Letter · Decision Letter 0]

4 Jan 2023

PONE-D-22-31261Capturing Epidemic Spread and Interplay with Multi-hop Contact Tracing InterventionPLOS ONE

Dear Dr. KIM,

Thank you for submitting your manuscript to PLOS ONE. After careful consideration, we feel that it has merit but does not fully meet PLOS ONE’s publication criteria as it currently stands. Therefore, we invite you to submit a revised version of the manuscript that addresses the points raised during the review process.

 Even though one reviewer recommends "minor revision", please consider both reviews as equally significant and deserving major changes in the paper, both in terms of results/analysis and presentation. 

We look forward to receiving your revised manuscript.

Kind regards,

Constantine Dovrolis

Academic Editor

PLOS ONE

Journal Requirements:

Reviewers' comments:

Reviewer's Responses to Questions

**Comments to the Author**

1. Is the manuscript technically sound, and do the data support the conclusions?

Reviewer #1: Yes

Reviewer #2: Yes

2. Has the statistical analysis been performed appropriately and rigorously? 

Reviewer #1: Yes

Reviewer #2: Yes

3. Have the authors made all data underlying the findings in their manuscript fully available?

Reviewer #1: Yes

Reviewer #2: Yes

4. Is the manuscript presented in an intelligible fashion and written in standard English?

Reviewer #1: Yes

Reviewer #2: Yes

5. Review Comments to the Author

Reviewer #1: Verdict

Major Revision

Summary

In this work, the authors formulate a mathematical model for the multi-hop contact tracing scheme that combines both the contact tracing and transmission dynamics in one unified framework using a Microscopic Markov Chain approach. They show that their MMCA model closely approximates the Monte Carlo simulations on both real and synthetic Watts-Strogatz network. In these simulations, the authors show that multi-hop contact tracing captures significantly more cases than the traditional one hop approach. Moreover, the authors show that the number of quarantined individuals is also significantly lower than in the traditional one hop approaches.

Relevance

The research problem is relevant and important for both policy makers and those in the academic community.

Novelty

Multi-hop contact tracing is not a new idea and has been well explored. It was used in the early HIV AIDS outbreak to perform contact tracing. Other settings of the broader idea behind multi-hop contact tracing are the friendship paradox and the scale-free nature of real life networks. However, to the reviewer’s best knowledge, this is the first systematic and mathematical approach to both model the spread of a viral disease like Covid-19 and the multi-hop contact tracing approach in one unified framework.

Strengths

1) Examines a very relevant research problem to both policymakers and the broader epidemiology community

2) Well formulated mathematical model that elegantly combines both multi-hop contact tracing and disease transmission.

3) Solid experimental results that clearly show the utility of the proposed model compared to baseline methods

4) MMCA approach is both close enough to MC simulations and scales up significantly better.

Weaknesses

Major Weaknesses

1) Watts-Strogatz networks are the only synthetic networks chosen. Though Watts-Strogatz networks have similar average path lengths and clustering coefficients, their degree distributions are unrealistic and fail to produce the classic hub and spoke structure that real life networks have. Consequently, the reviewer urges the authors to additionally add Barabasi-Albert networks, which possess such hub and spoke property, to their experimental test bench. This would strengthen the authors’ argument.

2) The entire work assumes that testing is ‘free’ and does not cost much. Though In Section III B 2., the authors argue that mass quarantining is much more expensive than mass testing, which the reviewer agrees with, the general assumption throughout the work is that testing capacity is virtually unlimited and easily scaled up which is unrealistic. Once the number of infections reaches critical mass, one is essentially testing the entire network which is not feasible. In the reviewer’s opinion, this is the single biggest weakness that holds back this paper. The main idea behind this work is that multi-hop contact tracing is better than single hop contact tracing. To show that, a valid comparison between the two approaches needs to be done. This requires the number of tests in both cases to be roughly equal or otherwise it is trivially obvious that the one that uses more tests will do better. To address this issue, the reviewer suggests two approaches. Any, or preferably both of these approaches, if included would significantly strengthen this paper.

a) Compare the multi-hop approach to a baseline that uses a similar number of tests for an accurate comparison. For eg., you could compare the 2-hop approach with a 1-hop approach and random tests to make up the difference for the 1-hope approach. A few more less trivial procedures where the multi-hop approach performs better would strengthen this paper.

b) Set a budget for the number of tests that is quite small compared to the size of the network and compare the performance of 1-hop and multi-hop approaches given that budget. This is very similar to the minimal sensor set on networks problem that has been well explored for both epidemic forecasting and other applications.

Minor Weaknesses

1) It is difficult to interpret the figures. Though the results are clearly positive, it is not clear how to interpret the figures present. For eg., in Figures 2 and 3 it is not immediately clear which network the experiment was run on. Additionally, in Figure 7, it is not clear which figure is better at first glance. The reviewer suggests that the authors improve the figure descriptions and the figures themselves for easier reading.

2) The notation when reintroduced late, like in Section III B 2., results in significantly harder reading. The reviewer was repeatedly turning back to look up the notation while reading through. The reviewer urges the author to simplify the notation and add more explanations for the model formulation in Section II for easier reading.

Reviewer #2: This is a very interesting paper that supports that 2-hop contact tracing is an effective strategy in reducing the number of infections as well as quarantine costs, using a microscopic Markov Chain approach, and I support its publication.

Here are some comments for improvement of the manuscript:

1. Many of the references refer to the coronavirus pandemic, and I have the impression that this paper is within this context as well. It might be beneficial to explicitly state that in the introduction and/or abstract title.

2. In the model formulation, 3rd paragraph: I would add a little bit more explanation to clarify what omega is.

Is it that a symptomatic infectious individual (I_s) gets tested with probability omega and when tested you assume that the test comes positive with probability 1?

Also, you have not discussed about false positive and false negative tests.If you make the assumption that tests are 100% accurate, then you should state this.

3. A diagram might be beneficial to the reader, showing the different states and arrows connecting them with the relevant probabilities. For example S -- L with +I_s β_I_s on top of the arrow, and two more arrows for β_I_p and β_I_a. Then under L you can have 1/λ to show the duration.

4. Typo in the 6th equation of the set of equations showing the evolution of probabilities (page 3): 2nd term of right hand side subscript of rho should be I_a.

5. For added clarity of eq.1 I would add some more explanation, eg prob of node i staying in S state is

1- Π_i^{S -> L} (t) = Π_{j=1} ^{N} prob of node i not getting infected via contact with node j

This will make the paper more accessible to readers not specialising in epidemic modelling, and set the basis for the rest of the paper.

In q.2 you need to state clearly that this is the one with no contact tracing.

6. Again I would add a small paragraph to explain eq. 3.

7. Page 5 paragraph starting with 'To verify': Referal to eq. (13) in line 2 and last line of that paragraph.

8. Page 5 paragraph starting with 'Next, we compute' you have the sentence:

'Individuals in symptomatic (Is) state can be tested and detected either after

showing symptoms or via contact tracing; thus, the probability that an individual in symptomatic (Is) state is tested

at time t is equivalent to the probability that the person is detected at time t.'

Here you are assuming 100% accuracy of test, right? It would be beneficial to state this.

9. Eq. 6 please add an explanation of the last term - why does omega appear there? This goes along with my second comment on clarifying what omega is.

10. Page 6: It might be beneficial to the reader to explain some terminology: Watts-Strogatz, average path lengths, clustering coefficients.

11. Page 7: Transmission probability beta now appears with no subscript. I guess you are assuming that beta_Is = beta_Ia = beta_Ip ? If so, please state it. Is this realistic?

12. Page 13 when discussing costs, what about the hospitalisation costs? Also, public health systems have limitations in hospitalisation capabilities, so relevant questions might be worth asking. I am not suggesting you add anything more to the current paper, but it might be worth discussing this in the discussion section.

13. When discussing the different strategies of no contact tracing, 1-hop or 2-hop contact tracing, it might be beneficial to consider the total number of contacts required to be traced per day. You do include the fraction f as an indication of a limitation to this, but in practice, wouldn't the contact tracing team have the limitation in the number of contacts traced per day, and not the fraction? Eg all the contacts can be traced at the beginning of the epidemic, but when it gets out of hand, the contacts of only 1000 detected people can be traced per day. I am not suggesting you add anything more to the current paper, but it might be worth discussing this in the discussion section.

6. PLOS authors have the option to publish the peer review history of their article (what does this mean?). If published, this will include your full peer review and any attached files.

Reviewer #1: No

Reviewer #2: No

---

## [Author Response · Author response to Decision Letter 0]

24 Mar 2023

We are grateful to the editors and reviewers for their time and valuable comments. We have made substantial revisions to our paper, and the manuscript has certainly benefited from these insightful comments and suggestions. Please find attached a point-by-point response explaining how we have addressed each of the reviewers’ comments. We hope that you find our responses satisfactory, and that the manuscript is now suitable for publication in PLOS ONE. We have marked in blue the parts of the manuscript that we have added, modified, or moved to a different location, in response to the comments in the reviews.

---

## [Decision Letter · Decision Letter 1]

26 Jun 2023

Capturing COVID-19 Spread and Interplay with Multi-hop Contact Tracing Intervention

PONE-D-22-31261R1

Dear Dr. KIM,

We’re pleased to inform you that your manuscript has been judged scientifically suitable for publication and will be formally accepted for publication once it meets all outstanding technical requirements.

Kind regards,

Constantine Dovrolis

Academic Editor

PLOS ONE

Additional Editor Comments (optional):

Reviewers' comments:

Reviewer's Responses to Questions

**Comments to the Author**

1. If the authors have adequately addressed your comments raised in a previous round of review and you feel that this manuscript is now acceptable for publication, you may indicate that here to bypass the “Comments to the Author” section, enter your conflict of interest statement in the “Confidential to Editor” section, and submit your "Accept" recommendation.

Reviewer #1: All comments have been addressed

Reviewer #2: All comments have been addressed

2. Is the manuscript technically sound, and do the data support the conclusions?

Reviewer #1: Yes

Reviewer #2: Yes

3. Has the statistical analysis been performed appropriately and rigorously? 

Reviewer #1: Yes

Reviewer #2: Yes

4. Have the authors made all data underlying the findings in their manuscript fully available?

Reviewer #1: Yes

Reviewer #2: Yes

5. Is the manuscript presented in an intelligible fashion and written in standard English?

Reviewer #1: Yes

Reviewer #2: Yes

6. Review Comments to the Author

Reviewer #1: The changes the authors made in response to the reviewer(s) comments have significantly improved the quality of the paper and the reviewer feels that the paper passes the publication threshold comfortably.

I suggest that the authors use higher resolution images for the plots (perhaps with vector images instead of raster ones) as some plots are a bit blurry.

Reviewer #2: I am pleased with the revised version of the paper, as the authors have taken into account the comments of the first review, significantly increasing the strength and clarity of the manuscript.

In addition to the much clearer explanation of the model, they have significantly expanded the results section by including additional computations for scale-free network (generated by Barab´asi-Albert

model) to capture the existence of hubs with an excessive number of connections, as well as addressing test fatigue.

As such I now recommend the paper for publication.

Below I have a couple of very minor suggestions, that I consider as optional.

I would include the description of the networks used (currently in III.A),

as a section IIC Network Descriptions and have subsections: Data-driven and Synthetic Networks. The authors could also add a sentence or two to when they introduce the scale-free network (generated by Barab´asi-Albert model).

Page 11: Instead of listing discrepancies inline in the paragraphs, I would put them in a table for the ease of the reader.

Conclusion section: the authors could include a couple of sentences summarising the new (ie after the first review) results addressing eg stress fatigue.

7. PLOS authors have the option to publish the peer review history of their article (what does this mean?). If published, this will include your full peer review and any attached files.

Reviewer #1: **Yes: **Vivek Anand

Reviewer #2: **Yes: **Panayiota Katsamba

---

## [Editor Report · Acceptance letter]

5 Jul 2023

PONE-D-22-31261R1 

Capturing COVID-19 Spread and Interplay with Multi-hop Contact
Tracing Intervention 

Dear Dr. Kim:

I'm pleased to inform you that your manuscript has been deemed suitable for publication in PLOS ONE. Congratulations! Your manuscript is now with our production department. 

Kind regards, 

on behalf of

Dr. Constantine Dovrolis 

Academic Editor

PLOS ONE